# An evolutionary hotspot defines functional differences between CRYPTOCHROMES

Clark Rosensweig[1,6], Kimberly A. Reynolds[2,3], Peng Gao[1], Isara Laothamatas[1], Yongli Shan[1], Rama Ranganathan[2,3,4,7], Joseph S. Takahashi [1,5] & Carla B. Green[1]

Mammalian circadian clocks are driven by a transcription/translation feedback loop composed of positive regulators (CLOCK/BMAL1) and repressors (CRYPTOCHROME 1/2 (CRY1/2) and PER1/2). To understand the structural principles of regulation, we used evolutionary sequence analysis to identify co-evolving residues within the CRY/PHL protein family. Here we report the identification of an ancestral secondary cofactor-binding pocket as an interface in repressive CRYs, mediating regulation through direct interaction with CLOCK and BMAL1. Mutations weakening binding between CLOCK/BMAL1 and CRY1 lead to acceleration of the clock, suggesting that subtle sequence divergences at this site can modulate clock function. Divergence between CRY1 and CRY2 at this site results in distinct periodic output. Weaker interactions between CRY2 and CLOCK/BMAL1 at this pocket are strengthened by co-expression of PER2, suggesting that PER expression limits the length of the repressive phase in CRY2-driven rhythms. Overall, this work provides a model for the mechanism and evolutionary variation of clock regulatory mechanisms.

[1] Department of Neuroscience, University of Texas Southwestern Medical Center, 5323 Harry Hines Boulevard, Dallas, TX 75390, USA. [2] Department of Biophysics, University of Texas Southwestern Medical Center, 6001 Forest Park Road, Dallas, TX 75390, USA. [3] The Green Center for Systems Biology, University of Texas Southwestern Medical Center, 6001 Forest Park Road, Dallas, TX 75390, USA. [4] Department of Pharmacology, University of Texas Southwestern Medical Center, 6001 Forest Park Road, Dallas, TX 75390, USA. [5] Howard Hughes Medical Institute, University of Texas Southwestern Medical Center, Dallas, TX 75390, USA. [6] Present address: Department of Neurobiology, Northwestern University, 2205 Tech Drive, Pancoe 2230, Evanston, IL 60208, USA. [7] Present address: The Center for the Physics of Evolving Systems, Biochemistry and Molecular Biology, The Institute for Molecular Engineering, University of Chicago, 929 East 57th Street, Chicago, IL 60637, USA. Correspondence and requests for materials should be addressed to C.B.G. (email: carla.green@utsouthwestern.edu)

ell autonomous molecular clocks with robust 24-h rhythms have evolved to coordinate physiological processes with daily changes in the environment. At the molecular level, clocks appear to have evolved independently a number of times and are comprised of a variety of different components[1]. In eukaryotes, these clocks are made up of transcription/translation feedback loops (TTFLs) with regulatory steps at many levels that tune the period and provide environmental input into the system.

The CRYPTOCHROMES (CRYs) play a critical regulatory role in clock function for both animals and plants. Though the precise molecular implementation of the TTFL varies across species, the mammalian circadian clock provides a clear illustration of how CRY functions. In this system, the TTFL is principally defined by a heterodimeric transcription factor, composed of basic helix–loop–helix (bHLH) PER-ARNT-SIM (PAS) domain-containing proteins CLOCK and BMAL1[2,3]. This transcription factor regulates the expression of many targets including the genes encoding the repressors: *Cry1/2* and *Per1/2/3*[4]. CRY and PER proteins translocate to the nucleus where they bind and repress CLOCK and BMAL1 activity[4,5]. Both CRY and PER are necessary to maintain rhythmicity in both cellular and organismal milieus[6–9], and it is known that degradation of the repressors PER and CRY is a critical period-determining step[10–15].

Several groups have begun to identify the functional interfaces on CRY involved in specific protein–protein interactions with CLOCK and BMAL1. In particular, BMAL1's transactivation domain (TAD) has been shown to interact with a C-terminal portion of CRY[16–18] and CRY's secondary pocket has recently been reported to mediate an interaction with CLOCK's PAS-B domain[19,20]. Notably, mutation of the secondary pocket residue R109 to a glutamine has been shown to reduce CRY1's repressive capacity and abrogate binding between CRY1 and CLOCK without disrupting the global CRY1 fold[19–21]. Additionally, it was recently reported that a human mutation that truncates CRY1's C-terminal tail strengthens the interaction between CRY1 and the CLOCK/BMAL1 heterodimer while also lengthening period in carriers[22]. Finally, crystallographic data demonstrates that PER2 and FBXL3, part of an E3 ubiquitin ligase complex that targets CRY for degradation, share an overlapping interface on CRY that includes an ancestrally important active site, a flavin-adenine dinucleotide (FAD)-binding pocket[20,23,24]. Despite these reports, it has been difficult to develop clear structure–function relationships for the wealth of functionally relevant mutations in CRY that have been reported[20,21,25–28] aside from mutations that have an effect on stability[29–32].

CRYs from all organisms share a stereotyped structure of a conserved Photolyase (PHL)-homology region, a C-terminal coiled-coil-like helix (CC helix), and a highly variable C-terminal extension known as the tail. However, they have evolved to perform a diversity of functional roles in the clock[33]. Thus they provide an excellent model system for understanding how subtle variation in sequence and structure can give rise to regulatory diversity and functional differences in clock timing. For example, the animal CRYs are broadly classified into types I and II. Only type II CRYs (such as CRY1 and CRY2 in mice) function as direct repressors of their transcriptional activators[34], whereas type I CRYs, such as the single CRY in *Drosophila* (dCRY), function as blue light-sensitive inputs to a TTFL composed of CLOCK and BMAL1 homologs and the repressors PER and TIMELESS[35]. Understanding how sequence variation in the type I and II CRYs leads to distinct functional roles thus provides one avenue to identify structural regions involved in differential regulation.

Within the type II CRYs, individual family members show further functional differences. The mammalian CRYs—here represented by mouse CRY1 and CRY2—have distinct functions

despite high sequence identity and similarity. First, CRY1 is a stronger repressor of CLOCK/BMAL1-mediated transcriptional activation than CRY2[27,36]. Second, although deletion of both CRYs results in arrhythmicity, individual null mutations result in opposite phenotypes—short ($mCry1^{-/-}$) and long ($mCry2^{-/-}$) periods, respectively[8,9]. Moreover, the CRYs are associated with a highly divergent set of DNA-binding complexes within the genome and a substantially different phase of peak binding[37]. The peak of occupancy for CRY2 is early in the evening in phase with PER1 and PER2, whereas CRY1 occupancy peaks in the late night and early morning, several hours after the peak occupancy of PER1 and PER2[37]. The sequence determinants of the regulatory differences between CRY1 and CRY2 are poorly understood. Overall, greater insight into the mechanisms governing how CRY determines periodicity is necessary for targeted development of therapies for circadian disorders.

Thus, major open questions include: (1) how do certain mutations in CRY lead to acceleration or deceleration of the clock; (2) what residues are involved in functional interfaces on CRY that mediate specific protein–protein interactions; (3) how are functional differences between type I and II CRYs manifested at a structural level; and (4) how do the structural features that differentiate CRY1 and CRY2 manifest divergent functional characteristics? To understand the determinants for CRY-mediated regulation of CLOCK/BMAL1 and for functional variation between the CRYs, we studied the pattern of amino acid conservation and co-evolution across the CRY/PHL family (CPF). This analysis reveals a co-evolving network of amino acids spanning the primary and secondary pockets that is conserved across both the CRYs and PHLs, indicating a shared family-wide functional architecture. We show the secondary pocket gates interactions with CLOCK and BMAL1 and likely differentiates type I and type II CRYs in their ability to directly repress CLOCK and BMAL1. Furthermore, we show that subtle changes at this pocket result in substantial differences in periodicity in cycling cells due to changes in affinity between CRY1 and CLOCK/BMAL1. Finally, we observe that a few small differences between CRY1 and CRY2 at this surface underlie CRY2's periodicity characteristics, again the result of a weakened affinity for CLOCK/BMAL1. Overall, these results define a mode of period regulation of the circadian clock that is orthogonal to the canonical model of period regulation arising from the stability of the PER and CRY repressor proteins.

## Results

**Co-evolving networks link the cavities of CRYS and PHLs**. To better understand the sequence determinants of CRY function, we conducted a statistical study of conservation and co-evolution across the CRYs and the PHLs, an evolutionarily related family of photoactivatable enzymes that catalyze the repair of ultraviolet-induced DNA lesions (Fig. 1a, b). PHLs have two cofactors: (1) an FAD molecule that binds in a well-conserved primary pocket (Fig. 1c) that directly interacts with and repairs the DNA lesion, and (2) a variable secondary cofactor (methenyltetrahyrdofolate or 8-hydroxy-7,8-didemethyl-5-deazariboflavin) that binds at a less conserved surface pocket (Fig. 1c) and that functions as a light-harvesting antenna[38]. The FAD molecule is essential for enzymatic activity, while the secondary cofactor modulates the dynamics of the reaction[39]. Interestingly, in a previous targeted mutational screen, we identified several positions in CRY1 and CRY2 that weaken repressive capacity and localize to the region on CRY structurally analogous to the PHL secondary pocket ((E103 (E121 in CRY2), G106, and R109), Fig. 1b). These results suggest that the PHL secondary pocket may have been repurposed in the CRYs for interactions with CLOCK/BMAL1.

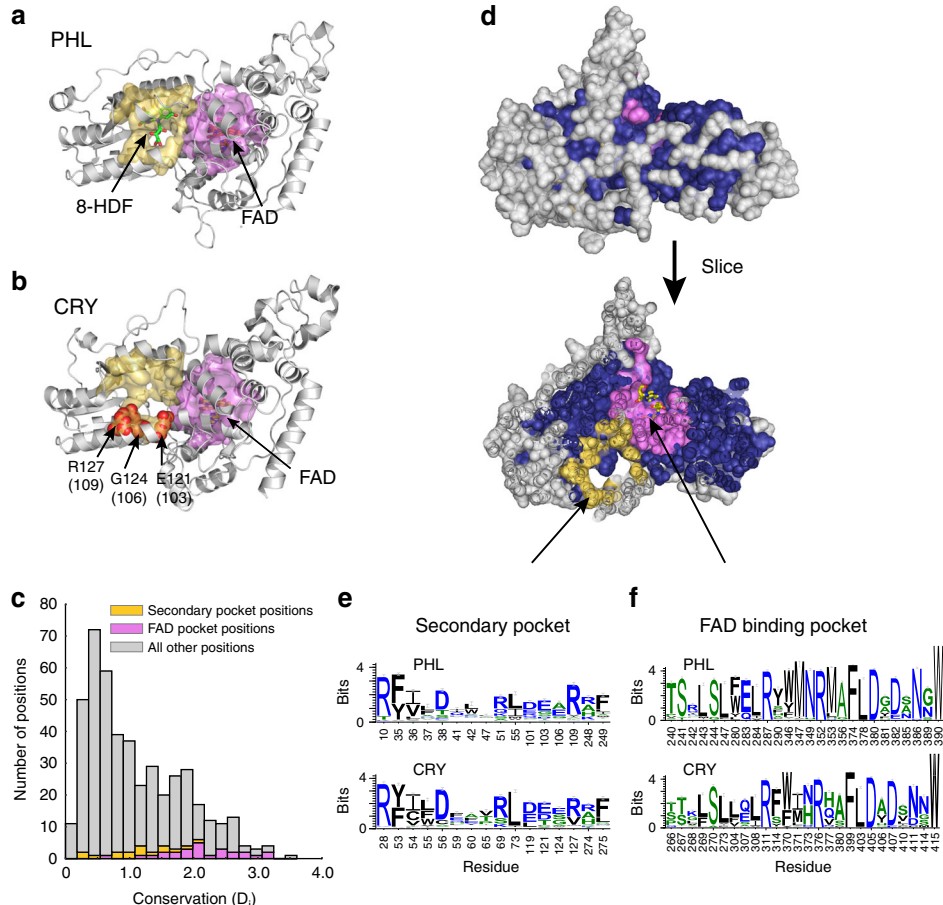

**Fig. 1** Identification of an allosteric secondary pocket in the CRYs by comparison to the PHLs. **a** The PHL domain is shown in gray cartoon (PDB: 1TEZ). The secondary pocket (yellow surface) and primary FAD-binding pocket (pink surface) were defined as all residues with at least one atom within 4 Å of 8-hydroxy-7,8-didemethyl-5-deazariboflavin (8-HDF) (green sticks) or FAD (yellow sticks) respectively. **b** The FAD-binding and secondary pockets mapped to the CRY2 structure (PDB: 4I6G). Three positions within the secondary pocket were previously identified in a functional screen (highlighted as red spheres). **c** Distribution of conservation values for all positions in the CRY/photolyase family (CPF) alignment. Conservation is computed as the Kullback–Leibler relative entropy ($D_i$), which measures the divergence of amino acid frequencies at a particular position from the distribution expected by random chance. Values near zero indicate that the observed frequencies at a site are close to random expectation, while values above three reflect positions approaching near-total conservation (84% identity or more). The majority of positions are weakly conserved; the secondary pocket positions (yellow) are moderately conserved, and the FAD-binding pocket positions (pink) are more conserved. **d** Mapping of the sector (dark blue spheres) to the mCRY2 structure. Non-sector positions are shown in gray spheres, and the primary and secondary pockets are again indicated in pink and yellow spheres respectively. **e–f** Sequence logo plots describing the sequence variation in the FAD-binding pocket and secondary pocket. Residue numbers correspond to the structures shown in **a** and **b**. The sequence variation is computed for sequences annotated as either CRYs or PHLs. We observe a similar degree of conservation within the CRYs and PHLs for both pockets

To investigate the possibility that functional coupling between the primary and secondary pockets is a conserved feature across both the CRYS and PHLs, we conducted the statistical coupling analysis (SCA), a method for inferring groups of conserved, co-evolving positions associated with complex functions such as allostery in a protein family[40,41]. These co-evolving groups are called sectors, and typically comprise physically contiguous networks of amino acids that link active sites to distantly positioned surfaces. Sectors correspond to allosteric mechanisms in several model systems, and recent work has shown that sectors create hotspots for allosteric regulation, enabling the evolution of new regulation in proteins through local modifications[42,43]. While distinct co-evolution-based approaches have also been developed to identify direct structural contacts in proteins (e.g., DCA, PSICOV, and GREMLIN), SCA uniquely focuses on discovering the collective networks of residues that underlie long-range coupling within a protein.

A key feature of SCA is the ability to mathematically relate the amino acid motif within sectors (correlated evolution of positions) to phylogenetic and/or functional divergences in the protein family (correlated evolution of sequences)[44,45]. This relationship permits us to test whether an allosteric mechanism inferred by SCA is conserved or is divergent in members of the protein family. For example, the family of Hsp70-like molecular chaperones contains both allosteric and non-allosteric members, and sector positions display the property of (1) cleanly separating these functional subfamilies and (2) displaying lower conservation in the non-allosteric group[45]. In contrast, the finding that sector positions do not separate subfamilies by function and do not show differences in overall conservation in subfamilies would be consistent with retention of an allosteric mechanism or a conserved functional coupling between sites. For the CPF, we expect SCA to reveal a sector linking the FAD-binding pocket (or primary pocket) with the secondary pocket. If so, we can then use

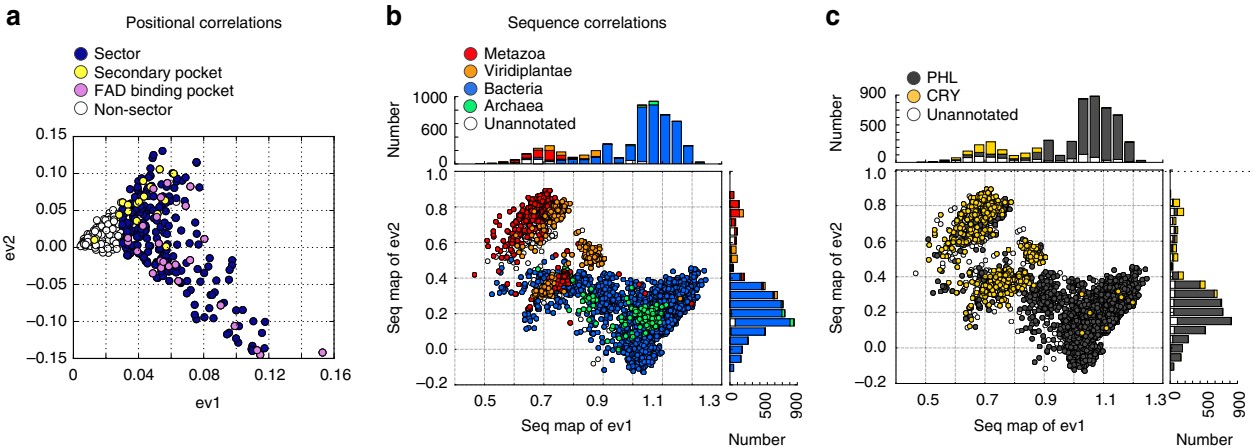

**Fig. 2** Statistical coupling analysis indicates that the allosteric secondary pocket is a conserved feature of CPF proteins. **a** Plot of the top two eigenvectors of the SCA correlation matrix. Each point represents an amino acid position, and amino acid positions that are proximal on this plot show similar patterns of co-evolution. The sector (blue points) is defined along eigenvector 1 (ev1). The secondary pocket positions and FAD-binding pocket positions (as defined in Fig. 1) are shown as yellow and pink dots, respectively. **b**, **c** Projection of the positional correlations to sequence divergences. In these plots, each point represents a CPF sequence. The sequences are color-coded either by phylogenetic annotation (**b**) or CRY/PHL annotation (**c**). Sequences which separate into groups along a particular axis diverge in the amino acid positions that separate along the corresponding axis in **a**

the sector positions to test the functional role of the secondary pocket in the CRY subfamily.

SCA for a multiple sequence alignment of 5385 CPF proteins reveals a single sector of 260 co-evolving amino acid positions that surrounds the primary FAD-binding site and extends to several surface accessible positions, including the secondary cofactor pocket (Fig. 1d and Supplementary Table 1). These findings are consistent with the known functional coupling between these sites in the PHLs. Interestingly, sequence logos characterizing the distribution of amino acids in sector positions comprising the primary and secondary pockets in PHLs and CRYs show similar patterns of positional conservation (Fig. 1e, f). This finding provides one level of support for the maintenance of functional coupling between the two pockets in these functional subfamilies.

To examine this more deeply, we plotted the top eigenvectors of the SCA co-evolution matrix and the corresponding projected sequence divergences. Sector residues emerge along the first eigenmode and are split by the second eigenmode as the first levels of a hierarchy of coevolution (Fig. 2a). Residues comprising the primary and secondary pockets occur diagonally in different directions along the top two modes. The corresponding sequence divergences (Figure 2b, c) show clear separations of groups of sequences in these directions, indicating variations of sector mechanism along different modes of sequence variation. However, we found that the sequence divergences dictated by the sector positions do not obviously correspond to phylogenetic groups in the alignment (Fig. 2b) or to the divergence of CRY and PHL sequences (Fig. 2c). Indeed, both phylogenetic classes and CRY/PHLs are mixed in all the sequence clusters defined by primary and secondary pocket positions. These findings support the model that the secondary pocket is likely a conserved functional feature of both PHL and CRY proteins. The associated sequence divergences provide new hypotheses for the variations of primary and secondary pocket function—a matter for further study.

We mapped the network of residues identified by SCA onto structures of the CRY2/PER2 CRY-binding domain (CBD) complex (Supplementary Fig. 1a) and the CRY2/FBXL3 complex (Supplementary Fig. 1b)[20,24]. Consistent with repurposing of the PHL active site in the CRYs, we found that PER2 and FBXL3 interact extensively with sector positions near the FAD-binding

pocket (where DNA lesions directly interact with PHLs) and around CRY2's CC helix. A logical hypothesis is that the secondary pocket has also been repurposed for regulatory interactions with CLOCK/BMAL1, with the residues in the yellow subgroup representing candidates for specifically contributing to the allosteric mechanism.

In summary, SCA exposes a co-evolving network of amino acids consistent with the known mechanism in the PHLs, and provides evidence for the conservation of functional coupling between the primary and secondary pockets in the CRY proteins as well. Since SCA provides residue-level resolution, we also have a logical basis for interrogating specific amino acid positions that are predicted to contribute to the allosteric mechanism.

**The secondary pocket is a binding interface for CLOCK/BMAL1.** In order to test residues identified in the secondary pocket in the SCA for a role in regulation of the clock mechanism, we first focused on the α4 helix between E103 and R109 of CRY1, where many (E103, P104, F105, G106, R109), but not all (K107, E108), amino acids were identified in the SCA (Fig. 3a). To test whether changes in these residues alter CRY1's function in the circadian clock, we used a rescue assay originally developed by Ukai-Tadenuma et al[46]. When expressed under the control of its endogenous promoter and an intronic element, $mCry1$ can rescue rhythms in a bioluminescent reporter ($Luciferase$($Luc$) driven by a $Per2$ promoter) in $Cry1^{-/-}/Cry2^{-/-}$ mouse embryonic fibroblasts (Fig. 3b). We tested two mutations identified in the screen described above (E103K and G106R) and an additional mutant chosen to be less disruptive of the local protein environment (F105A) due to the more conservative mutation and the solvent-exposed nature of the wild-type (WT) residue. We found that none of the mutants were able to rescue rhythmicity, resulting in high constitutive LUC activity indicative of unchecked transcriptional activity by CLOCK and BMAL1 (Fig. 3c). In addition, all three mutants had severe deficits in their ability to co-immunoprecipitate (co-IP) the CLOCK/BMAL1 heterodimer, but bound PER2 comparably to WT CRY1 (Fig. 3d and Supplementary Fig. 2). Since PER2 adopts an extended binding interface with CRY1 and CRY2[20,23], these data suggest that mutations at the "lower" (relative to the orientation shown here) interface (α4) of the secondary pocket cause local disruptions in CRY1's structure rather than global disruption of its protein fold.

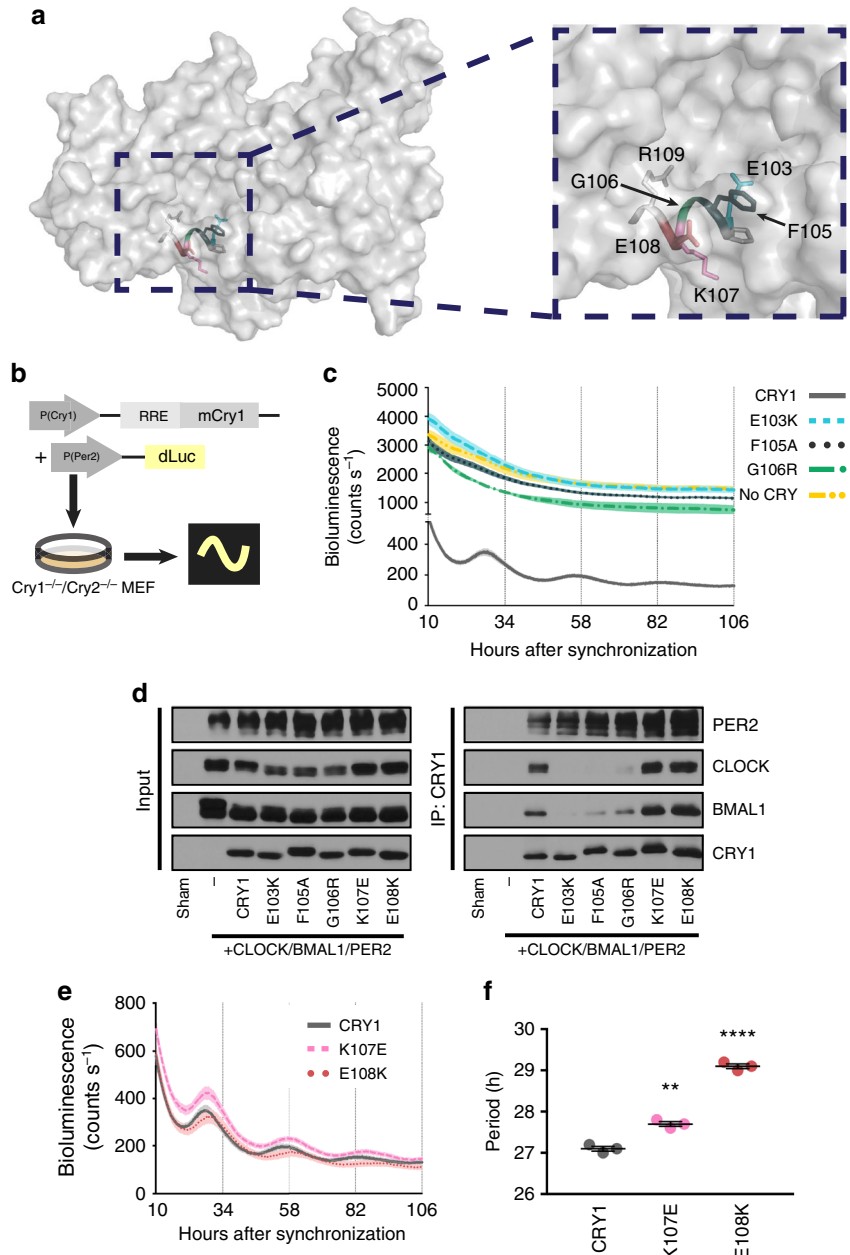

**Fig. 3** The lower helical boundary of CRY's secondary pocket gates interaction with CLOCK. **a** A surface view of the structure of CRY1 (PDB: 5T5X) with the lower helix of the secondary pocket shown as a ribbon diagram with side chains shown as sticks and relevant residues color-coded for the rest of the figure. Inset is a magnified view of this helix with residues labeled. **b** Schematic of *Cry1* rescue assay. RRE is an ROR response element necessary for delayed expression of CRY. **c** Rescue assays with CRY1 mutants identified in the SCA ($n = 3$ per condition, reflective of 6 plates from two independent experiments) shown as means ± SEM. **d** Co-IP assay with PER2, CLOCK, BMAL1, and various CRY1 mutants. Multiple bands in PER2, CLOCK, and BMAL1 lanes are indicative of post-translational modifications on these proteins. Blot is representative of at least three independent experiments. **e** Rescue assays for two CRY1 residues not identified in the SCA ($n = 3$ per condition, reflective of 6 plates from two independent experiments) shown as means ± SEM. **f** Period plot for the data shown in **e**. Asterisks show significance by unpaired $t$ test with Welch's correction (**$p = 0.0018$, ****$p < 0.0001$)

In contrast, even severe mutations of the non-sector residues at positions 107 (K107E) and 108 (E108K) of CRY1 did not impair CRY1's ability to rescue rhythms (Fig. 3e, f), and these mutants interacted strongly with CLOCK, BMAL1, and PER2 (Fig. 3d and Supplementary Figs. 2 and 10a), suggesting that these residues are not critical for binding to CLOCK and BMAL1. Consistent with this interpretation, K107 and E108 are superficial residues with side chains that neither gate access nor form any part of the surface of the interior of the pocket. Based on these results, we suggest that the interior of the secondary pocket is a critical interface for interactions between CRYs and their repressive targets CLOCK and BMAL1, consistent with another recent report[19]. Ultimately, in concert with this report, we conclude that the secondary pocket is a site for interaction with CLOCK specifically. These results also support the predictions from SCA that a subset of residues in the secondary pocket primarily mediate regulation of this protein–protein interaction.

**Insect and vertebrate CRYs diverge at the secondary pocket.** The two types of ancestral animal CRYs (type I and type II) can be found in various combinations in different animals. For

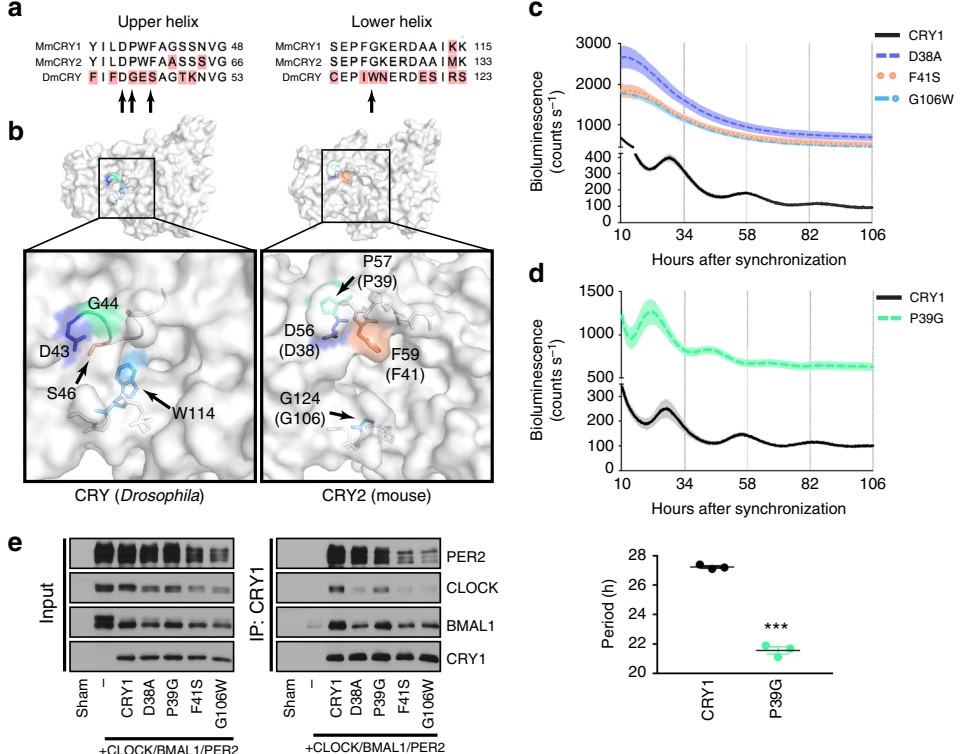

**Fig. 4** Structural differences at the secondary pocket differentiate type I and type II CRYs functionally. **a** Alignment of the amino acids in the upper and lower helical boundaries of the secondary pocket of CRYs from *Mus musculus* (Mm) and *Drosophila melanogaster* (Dm). Residues examined in Fig. 4 indicated by arrows. Red shading indicates disagreement with consensus. **b** Surface view of the secondary pocket of dCRY (PDB: 4GU5) and CRY2 (PDB: 4I6E). Orthologous residues in dCRY and mCRY2 identified in the SCA and examined in greater detail here are colored here in correspondence with the rest of the figure. CRY1 residues are shown in parentheses. **c** Rescue assays for mutants of three residues in CRY1 identified in the SCA (*n* = 3 per condition, reflective of 9 (D38A), 12 (F41S), and 9 (G106W) plates from 3–4 independent experiments) shown as mean ± SEM. **d** Rescue assay for CRY1 P39G (*n* = 3, reflective of 6 total plates from two independent experiments) shown as mean ± SEM. Period plot for rescues in **d** shown in the plot below. Mean ± SEM indicated by bars. Asterisks show significance by unpaired *t* test with Welch's correction (***$p$ = 0.0006). **e** Co-IP assay with PER2, CLOCK, BMAL1, and various CRY1 mutants. Blot is representative of at least three independent experiments

example, some insects have only a type I CRY (*Drosophila*), some have only a type II CRY (the honey bee, *Apis mellifera*), and some have both (the monarch butterfly, *Danaus plexippus*)[34,47]. We hypothesized that an evolving interface for direct interaction between CRY and core clock components might underlie this evolutionary divergence. Our finding that the secondary pocket is an interface for binding between CLOCK and CRY led us to look carefully at other residues in the secondary pocket identified in the SCA. In addition to the "lower" helical boundary of the pocket discussed above (α4), a subset of residues was identified in a flexible loop forming the "upper" boundary of the pocket (located between β2 and α2 in CRY1). An alignment of the α4 helix residues and the "upper" loop residues from various vertebrate and insect sequences reveals divergence at both sites between the two groups (Fig. 4a and Supplementary Fig. 3). Moreover, a conserved glycine in the type II vertebrate-like CRY population (position 106 in CRY1) is a conserved tryptophan in the type I CRY population (Fig. 4a and Supplementary Fig. 3). Structural comparison between CRY2 (for which the "upper" loop has been solved) and dCRY reveals that this tryptophan protrudes into and fills much of the cavity of the secondary pocket in dCRY (Fig. 4b). Furthermore, in the absence of a ligand, a more structured "upper" loop in CRY2 causes the pocket to adopt an open and exposed conformation suitable for a potential protein–protein interaction. All type II CRY proteins examined retain the sequence features of the "upper" and "lower" boundaries of this pocket, suggesting that it might be a major site of differentiation between type I and II proteins.

We tested the hypothesis that differences at these two interfaces partially underlie the more direct role of CRY in vertebrate clocks. Single amino acid substitutions in CRY1 of the divergent residues at positions identified in the SCA (P39G, F41S, G106W) caused either weakly repressive, short period rescues (P39G), or arrhythmicity (F41S, G106W) (Fig. 4c, d). Substitution at a convergent residue (D38A) from the SCA network also resulted in no rescue (Fig. 4c). All of these substitutions resulted in attenuated interactions with CLOCK and BMAL1, though P39G retains the strongest interaction (Fig. 4e and Supplementary Fig. 4). Interactions with PER2 were mostly intact, though there was a trend toward a weakened interaction between PER2 and the F41S and G106W substitutions (Fig. 4e and Supplementary Figs. 4 and 10b). Taken together, these data suggest that the secondary pocket of type II CRYs has evolved a more accessible conformation through which CRY interacts with the CLOCK/BMAL1 heterodimer. This interaction defines a critical feature of vertebrate-like clocks and likely underlies, at least in part, the direct role of CRY in vertebrate-like clocks.

**The secondary pocket interface tunes periodicity.** In addition to the residues on the "upper" and "lower" boundaries of the secondary pocket, multiple residues in the "upper right corner" of the pocket were identified in the SCA network (Fig. 5a). We hypothesized that they might also play a role in binding to CLOCK and BMAL1, so we mutated the residues to alanines and tested them in our rescue assay. All three mutant CRY1s (R51A,

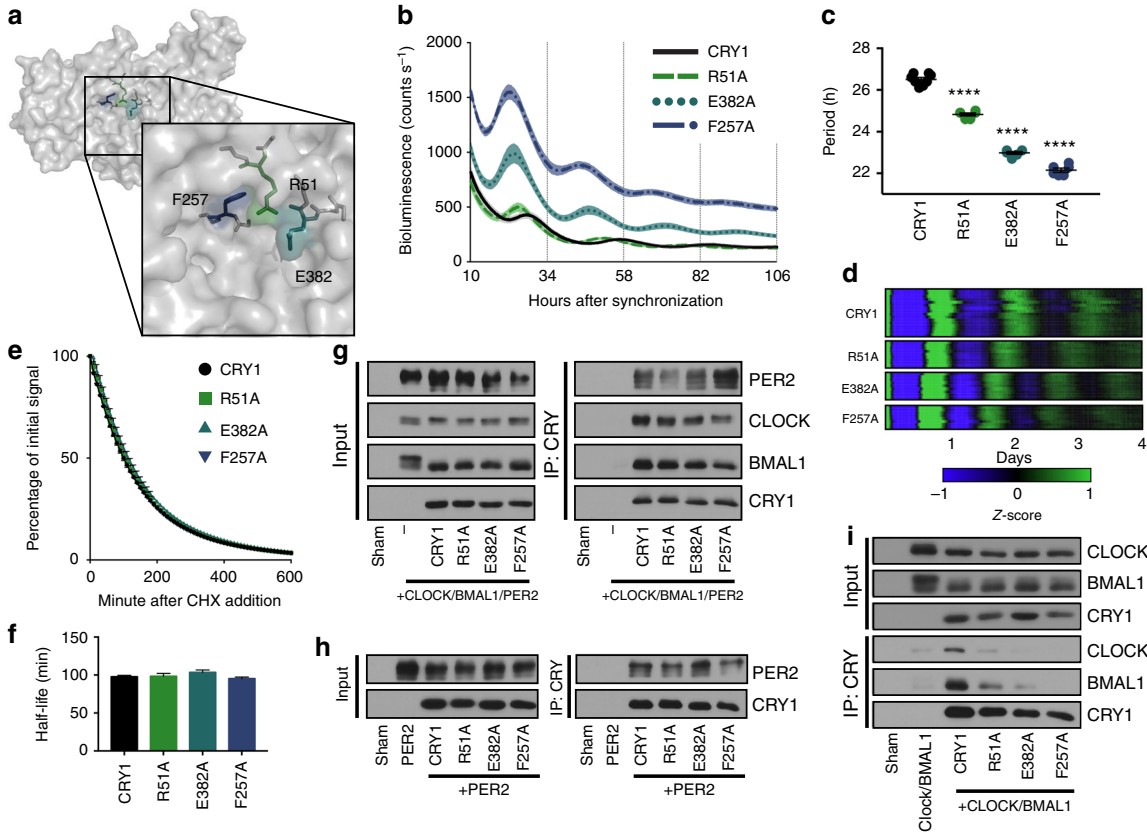

**Fig. 5** Weakened interaction between CRY1 and CLOCK/BMAL1 dramatically shortens the period in rescue assays. **a** Surface view of CRY1 (PDB: 5T5X) with an inset magnification of the secondary pocket. Residues R51, E382, and F257 are labeled and colored in green, teal, and navy blue. **b** Rescue assays performed with WT, R51A, E382A, and F257A CRY1 ($n = 8$, 6, 6, and 6 per condition, respectively, reflective of 9 (R51A), 15 (E382A), and 12 (F257A) plates from three to five independent experiments), shown as mean ± SEM. **c** Period plot for rescues shown in **b**. Mean ± SEM indicated by bars. Asterisks indicate significance by unpaired $t$ test with Welch's correction (****$p < 0.0001$). **d** Heat maps of CRY1 and mutant rescues demonstrate period and phase differences over multiple cycles. Raw data were baseline subtracted and $z$-scores were calculated, and then scaled to a range of −1 to 1. The data from 16 WT, 9 R51A, 9 E382A, and 8 F257A plates are shown. **e** Degradation assay with CRY1::LUC and mutants ($n = 3$ per condition). Samples were normalized to initial luminescence signal. Half-life was determined by fitting a one-phase decay curve to the data. **f** Half-lives from **e** shown as means + SEM. No significant difference between WT and mutants by unpaired $t$ test with Welch's correction (R51A: $p = 0.8245$; E382A: $p = 0.0660$; F257A: $p = 0.2029$) **g** Co-IP assay with PER2, CLOCK, BMAL1, and CRY1 mutants. Blot is representative of three independent experiments. **h** Co-IP assay with PER2 and CRY1 mutants. Blot is representative of three independent experiments. **i** Co-IP assay with CLOCK, BMAL1, and CRY1 mutants. Blot is representative of three independent experiments

E382A and F257A) rescued rhythms with dramatically shortened periods compared to the WT rescue (Fig. 5b–d). Moreover, the E382A and F257A rescues had higher amplitude rhythms and were less repressive overall as shown by their increased luminescence signal (Fig. 5b). Acceleration and deceleration of the clock have consistently been associated with changes in the degradation rate of the repressors, CRY and PER[14,15,29,48]. In order to test whether substitutions at these residues were causing a change in the degradation rate of CRY, we expressed a CRY1::LUC fusion protein in HEK-293A cells and treated with cycloheximide to block new protein synthesis, monitoring the decay in luminescence as a reporter for protein degradation. Introducing a mutation that stabilizes CRY1 (S588D)[29] led to deceleration of the rate of luminescence signal decay (Supplementary Fig. 5a, b). This finding is consistent with a recent report in which the half-lives of 36 CRY1 mutants, including S588D, were determined using a similar construct that faithfully reported a wide range of degradation rates[49]. However, none of the short period pocket mutants (R51A, F257A, E382A) affected the rate of decay (Fig. 5e, f).

We hypothesized that these period-shortening pocket mutations modulate the affinity between the CLOCK/BMAL1 heterodimer and CRY1. Though there was a trend towards weaker interaction, co-IP of CLOCK and BMAL1 in the presence of overexpressed PER2 was not significantly affected (Fig. 5g and Supplementary Figs. 5c and 10c) and none of the mutants showed significant changes in PER2 binding when overexpressed only with PER2 (Fig. 5h and Supplementary Figs. 5d and 10d). However, when overexpressed with just CLOCK and BMAL1, some of these mutants showed a significant decrease in interaction with CLOCK and BMAL1 in proportion to overall shortening of the rescue period (Fig. 5i and Supplementary Figs. 5e and 10e). These results imply that mutations in the secondary pocket can modulate the affinity between CRY1 and the CLOCK/BMAL1 heterodimer, which in turn tunes the period of the oscillation. This effect is most obvious without PER2, suggesting that the shortened period in these mutants might stem from an increasing requirement of PER co-expression to stabilize the repressive interaction of CRY with CLOCK and BMAL1.

In summary, these results argue that through regulation at a distance, variations in the secondary pocket can provide a fine-tuning of CRY function through modulation of CLOCK/BMAL1 binding.

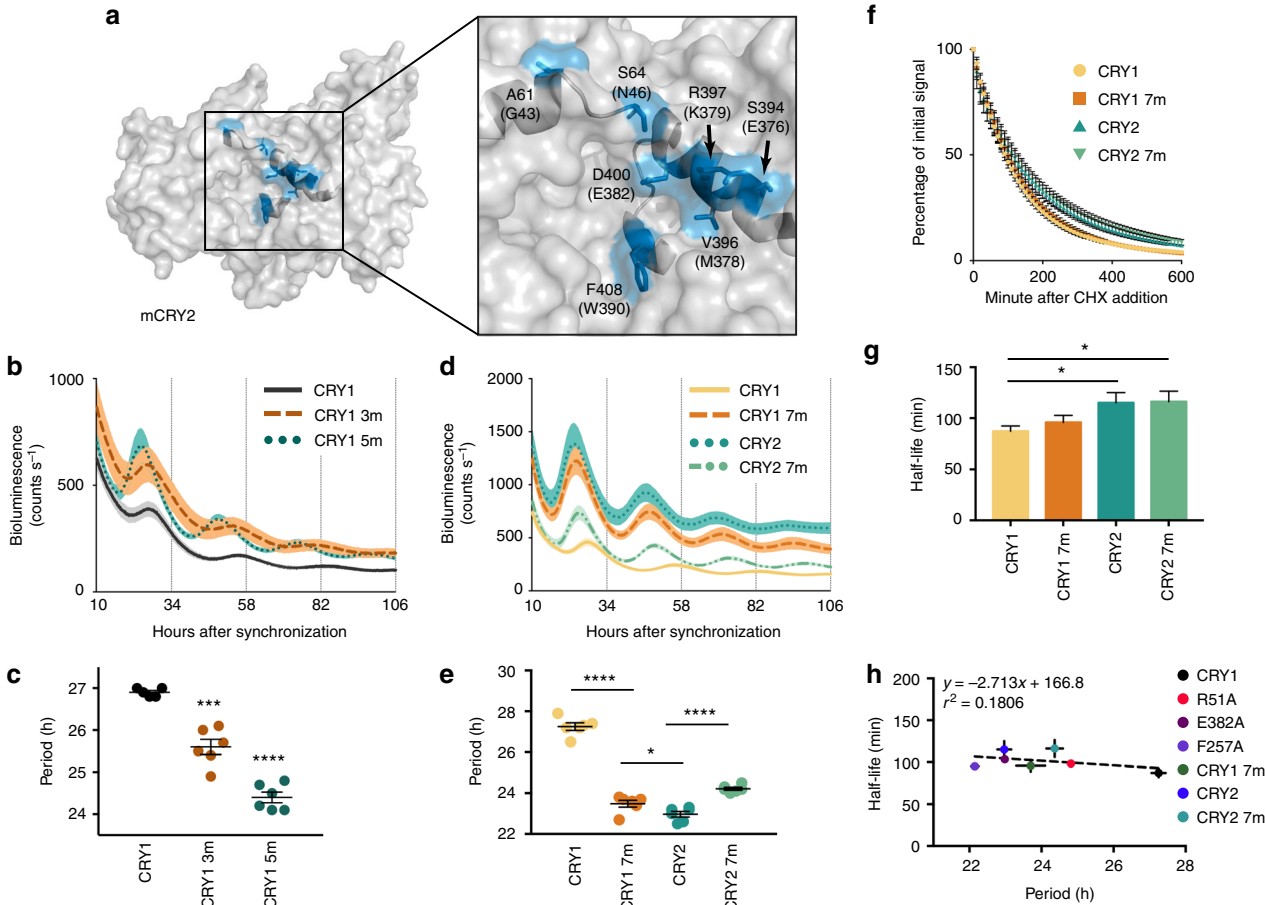

**Fig. 6** Subtle divergence between CRY1 and CRY2 at the secondary pocket largely dictates periodicity differences between the two repressors. **a** Surface view of the secondary pocket in CRY2 (PDB: 4I6E) with the seven divergent paralogous residues of CRY1 and CRY2 colored blue and labeled. CRY1 residues are shown in parentheses. **b** Rescue assays performed with WT CRY1, CRY1 3m (CRY1 M378V/K379R/E382D), and CRY1 5m (CRY1 E376S/ M378V/K379R/E382D/W390F) shown as mean ± SEM ($n = 5, 6, 6$ per condition, reflective of 18 plates each from 6 independent experiments). **c** Period plot of rescues in **b**. Mean ± SEM indicated by bars. Asterisks show significance by unpaired $t$ test with Welch's correction (*** $p = 0.0005$, **** $p < 0.0001$) compared to WT. **d** Rescue assays performed with WT CRY1, CRY1 7m (CRY1 G43A/N46S/E376S/M378V/K379R/E382D/W390F), WT CRY2, and CRY2 7m (CRY2 A61G/S64N/S394E/V396M/R397K/D400E/F408W) ($n = 6$ per condition, reflective of 30 (CRY1 7m), 33 (CRY2), and 24 (CRY2 7 m) plates from ≥8 independent experiments) shown as mean ± SEM. **e** Period plot of rescues in **d**. Mean ± SEM indicated by bars. Asterisks show significance by unpaired $t$ test with Welch's correction (* $p = 0.0387$, **** $p < 0.0001$). **f** Degradation assay with CRY1::LUC and CRY2::LUC and mutants ($n = 6$ per condition, 2 independent experiments). Samples were normalized to the initial luminescent signal. **g** Half-lives shown as mean ± SEM. Half-life was determined by fitting a one-phase decay curve to the data. Asterisks show significance by unpaired $t$ test with Welch's correction (* $p < 0.05$ (CRY2 vs. CRY1: $p = 0.0407$; CRY2 7 m vs. CRY1: $p = 0.0376$), NS (CRY1 7m vs. CRY1: $p = 0.3484$; CRY2 7m vs. CRY2: $p = 0.9348$; CRY2 vs. CRY1 7 m: $p = 0.1483$; CRY2 7m vs. CRY1 7 m: $p = 0.1331$)). **h** Correlation plot of period vs. half-life for the various pocket mutant rescues. Half-lives and periods shown as mean ± SEM. Linear regression is shown as a dotted line. The slope's deviation from zero was not significant ($p = 0.3418$)

**Subtle divergence underlies CRY1/2 periodicity differences.** One of the outstanding mysteries in circadian biology is the profound difference in periodicity of free-running rhythms between $Cry1^{-/-}$ and $Cry2^{-/-}$ mice. $Cry1^{-/-}$ mice have short endogenous periods (~22.5 h) and $Cry2^{-/-}$ mice have long endogenous periods (~24.6 h)[8,9]. Moreover, CRY1 functions as a stronger repressor of CLOCK/BMAL1-mediated transcriptional activation[27,36]. Using the $Cry1$ promoter to drive $Cry2$ expression in $Cry1^{-/-}/Cry2^{-/-}$ mouse embryonic fibroblasts, Khan et al.[27] reported that they were unable to rescue rhythms with WT $Cry2$. However, if they swapped residues 313–426 of CRY1 for the homologous residues in CRY2, this chimeric CRY2 was able to rescue rhythms. This experiment suggests that some set of differential residues within this domain is likely to play a critical role in driving repression of CLOCK/BMAL1-mediated transcriptional activation. There are only 12 residues that are different between CRY1 and CRY2 in this 113 amino acid domain.

Intriguingly, five of those residues (E376/S394, M378/V396, K379/R397, E382/D400, and W390/F408) are located superficially at the "right" boundary of the secondary pocket (Fig. 6a) and the amino acid changes are conservative. Moreover, three out of five of the residues were identified as part of the SCA network (M378/ V396, E382/D400, and W390/F408).

Given our finding that substitutions in this region of CRY1 can cause substantial shortening of the period of the circadian oscillation, we hypothesized that several subtle changes at this interface might underlie the shorter period in clocks driven only by CRY2. Converting any of these five residues in CRY1 individually to its CRY2 homolog had a minimal effect on the periodicity of the rescue (Supplementary Fig. 6a, b) with the greatest effect observed in E376S and W390F rescues, which were accelerated by roughly an hour per day (Supplementary Fig. 6b). However, combinatorial conversion of any of these residues to their CRY2 homologs led to increasingly short rescue periods,

including the three residues that had no effect at the individual level (M378V/K379R/E382D, i.e., CRY1 3 m) (Fig. 6b, c). Mutating all five residues in concert (CRY1 5m) led to the most substantial shortening compared to WT (24.4 h vs. 26.9 h) (Fig. 6c). Examination of the residues contributing to the surface area of the pocket shows that in addition to the five residues identified above, there are only two other divergent residues: G43/A61 and N46/S64 (Fig. 6a).

Mutation of both residues also led to an accelerated rescue rhythm compared to WT CRY1 (Supplementary Fig. 6c, d). Furthermore, combinatorial mutation of these residues also resulted in a corresponding weaker repression of CLOCK/BMAL1 transcriptional activation as demonstrated by the higher overall luminescence in these rescues (Supplementary Fig. 6c).

***Cry2 rescues rhythms with short periods similar to in vivo data.*** Khan et al.[27] demonstrated that even under the control of the *Cry1* promoter, *Cry2* was unable to rescue rhythms in *Cry*-deficient cells. We reasoned that if the seven residues at this pocket differentiate CRY1 and CRY2 at a functional level, then mutation of all seven together in CRY1 should cause it to fail to rescue rhythms. As a control for this experiment, full-length, WT *Cry2* was cloned into the *Cry1* rescue vector. However, we found that both the seven-residue pocket mutant (CRY1 7m) and CRY2 were competent to rescue rhythms (Fig. 6d). Moreover, both CRY1 7m and CRY2 produced rhythms with significantly shortened periods (23.48 h and 22.97 h, respectively) compared to WT CRY1 (27.25 h) (Fig. 6e), consistent with in vivo data from *Cry1*$^{-/-}$ mice, which also have short periods of ~22.5 h[8,9]. Compared to WT CRY1, both CRY2 and CRY1 7m were substantially derepressed and had higher amplitude oscillations. These results suggest that subtle divergence at this interface is a major driver of period differences in CRY1-driven or CRY2-driven rhythms. To test this hypothesis further, all seven residues in CRY2 were converted to their CRY1 homologs (denoted CRY2 7m). The effect of these mutations was again additive (Supplementary Fig. 6e, f), and resulted in a significant lengthening of the period (24.22 h) and an increase in repression compared to WT CRY2 (Fig. 6d, e), though these effects were more modest than the CRY1 7m mutant.

Given that we were using the same cell line as Khan et al.[27] and a nearly identical *Cry2* rescue construct, we were surprised to see a strong rescue by CRY2. Despite testing the few obvious variables that we could identify between these experiments, we ultimately could not clearly determine the reason underlying diverging reports of *Cry2*'s capacity to rescue rhythms[22,27,50].

To determine whether the pocket architecture affects the intrinsic stability of CRY1 and CRY2, we measured the half-life of CRY1, CRY2, CRY1 7m, and CRY2 7m, and found that, consistent with a previous report, CRY2 is more stable than CRY1[50] (Fig. 6f, g). However, neither CRY1 7m nor CRY2 7m differed significantly from CRY1 or CRY2, respectively, in half-life, suggesting that the period differences seen in the rescue assays were not due to changes in stability. Moreover, we found no correlation between half-life and period for any of the mutant residues in the secondary pocket that rescue rhythms (Fig. 6h). We also performed rescues with *Cry1*, *Cry2*, *Cry1 7m*, and *Cry2 7m* over a broad range of doses (25 ng to 1200 ng of rescue vector) and found a small effect of dose on period length (Supplementary Fig. 7). However, the results consistently show a difference between *Cry1* and the other rescues at all doses, suggesting that differences in DNA dosage are not driving the divergent periods.

**PER2 facilitates stable CRY2:CLOCK:BMAL1 complexes.** Since some of the mutations in the secondary pocket caused dramatic

acceleration of the clock by attenuating the interaction between CRY1 and CLOCK/BMAL1 (Fig. 5i), we tested whether this was also the case for the short period rhythms in the CRY2 rescues. When co-expressed with PER2, both CRY1 and CRY2 had a strong interaction with CLOCK and BMAL1, with no significant differences between the two (Fig. 7a and Supplementary Figs. 8a and 11a). CRY2 7m had a similarly robust interaction, but CRY1 7m demonstrated an attenuated interaction with CLOCK. There were no clear differences in interaction with PER2 when coimmunoprecipitated with or without overexpressed CLOCK and BMAL1 (Fig. 7a, b and Supplementary Figs. 8a, b and 11a, b). However, when co-IPs were performed without overexpressed PER2, both CRY1 7m and CRY2 displayed a weakened interaction with CLOCK and BMAL1 compared to WT CRY1 (Fig. 7c and Supplementary Figs. 8c and 11c). In contrast, CRY2 7m had a significantly stronger interaction with CLOCK and BMAL1 compared to CRY1 7m and CRY2 and a significantly stronger interaction with BMAL1 compared to CRY1 (Fig. 7c and Supplementary Figs. 8c and 11c). Together, these data suggest that in the context of other structural features, a CRY1-like pocket strengthens the interaction with CLOCK/BMAL1 when PER2 is not present. However, the presence of PER2 brings parity to the interaction of CRY1, CRY2, and CRY2 7m with CLOCK/BMAL1.

Several recent reports characterizing the interaction between CRY and CLOCK[19] and CRY and BMAL1[18] suggest that the secondary pocket is gating the interaction between the heterodimer primarily through a direct interaction with CLOCK. Taking into account these results along with our data suggesting that timing differences between CRY1 and CRY2 are primarily generated by divergence at the secondary pocket, we chose to focus on CLOCK rather than BMAL1 to test whether CRY1 has a stronger interaction than CRY2 with CLOCK/BMAL1 in a cellular milieu. We performed a reciprocal two-color bimolecular fluorescence complementation (BiFC) competition assay (Fig. 7d). In this assay, a C-terminal Cerulean (CerC) fragment can interact with either an N-terminal Venus (VenN) or N-terminal Cerulean (CerN) fragment to produce fluorescence in the Venus or Cerulean range, respectively. Non-complementary controls were used to establish background fluorescence as previously described[15]. A CerC-CLOCK construct containing the PAS-A and PAS-B domains was expressed with VenN-tagged and CerN-tagged CRYs and we found that CRY1, regardless of its tag, interacted with CLOCK more strongly than CRY2 (Fig. 7e, f), which suggests that the interactions seen in our co-IPs are relevant in vivo. These data strongly support the idea that the seven unique residues at the secondary pocket of CRY1 and CRY2 are critical for gating a strong physical interaction with CLOCK and BMAL1. They also suggest that expression of PER restricts the phase of repression for CRY2 by facilitating the formation of a stable repressive complex, while CRY1 is able to maintain a repressive complex even as PER levels decrease.

**CRY1's pocket and tail are necessary for long period rescues.** Converting CRY1's secondary pocket to a CRY2-like architecture was sufficient to diminish its interaction with CLOCK and BMAL1 and accelerate the clock speed by 4 h. However, despite greater gains in interaction with CLOCK and BMAL1, CRY2 7m had only a modest period lengthening effect compared to WT CRY2. The major region of structural divergence between CRY1 and CRY2 is the tail—a highly disordered region with no similarity between the two CRYs. Given that the secondary pocket does not fully account for periodicity differences, we hypothesized that the tail might play an additional role in determining periodicity. To test this hypothesis, we generated chimeric CRY constructs in which either the secondary pocket (CRY1 7m,

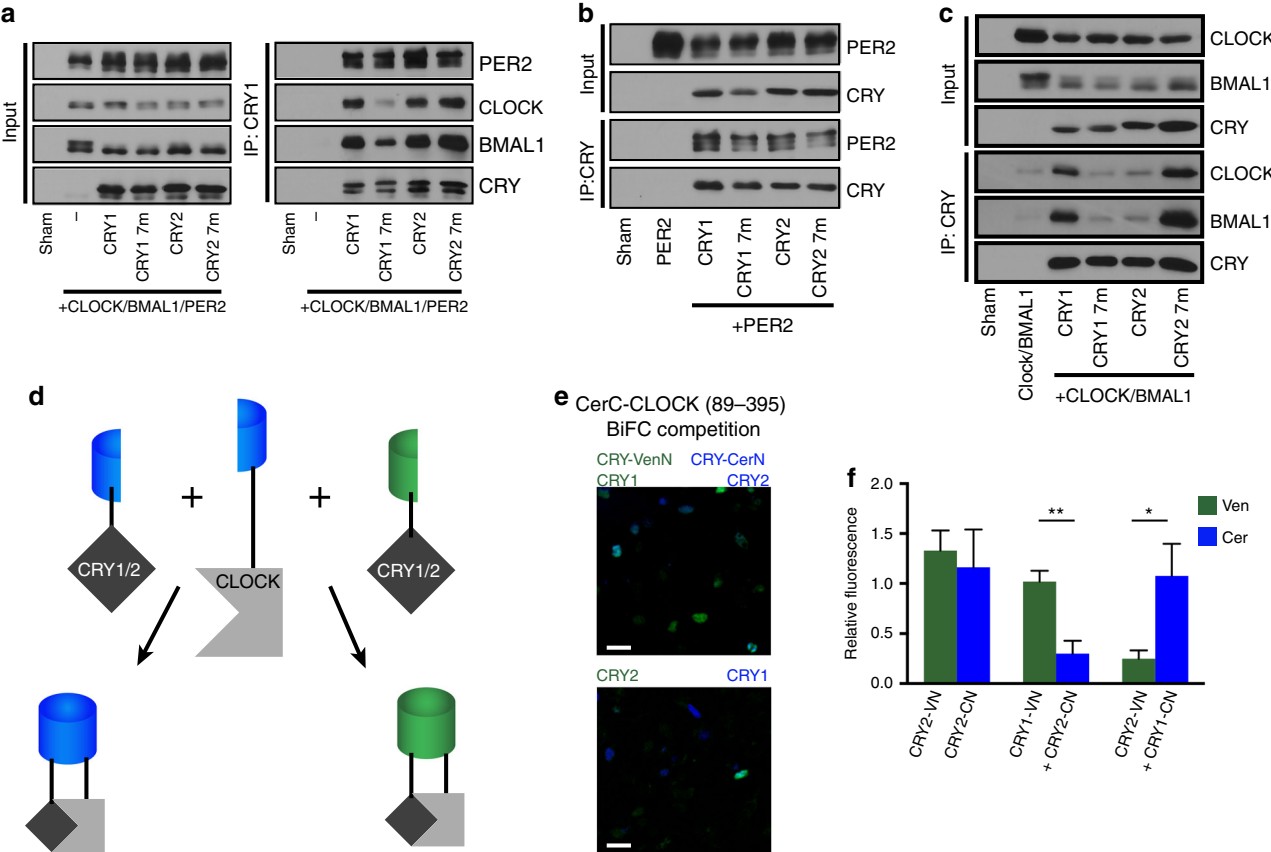

**Fig. 7** CRY2 requires PER to form a stable complex with mCLOCK and mBMAL1. **a** Co-IP assay with mPER2, mCLOCK, mBMAL1, and CRY1, CRY2, or pocket-switched mutants. Multiple bands in mPER2, mCLOCK, and mBMAL1 lanes are indicative of post-translational modifications on these proteins. The upper band is CRY, the lower band is a nonspecific band recognized by the V5 antibody used to probe for mPER2. Blot is representative of three independent experiments. **b** Co-IP assay with mPER2 and CRY1, CRY2, or pocket-switched mutants. Blot is representative of three independent experiments. **c** Co-IP assay with mCLOCK and mBMAL1, and CRY1, CRY2, or pocket-switched mutants. Blot is representative of three independent experiments. **d** Schematic of bimolecular fluorescence complementation competition assay. A CLOCK construct (residues 89–395) is N terminally tagged with a C-terminal fragment of Cerulean, which can interact with either an N-terminal fragment of Venus or Cerulean to produce yellow or blue fluorescence. CRY1 or CRY2 fused to these N-terminal fragments compete to bind CerC-CLOCK and fluorescence is used as a readout in the competition for binding. **e** Reciprocal two-color, three-way bimolecular fluorescence complementation (BiFC) in 293A cells using CerC-CLOCK (89–395) complementation with CRY1-VenN + CRY2-CerN or CRY1-CerN + CRY2-VenN. Pseudocoloring: Venus (Green), Cerulean (Blue). Scale bar is 30 μm. **f** Quantification of BiFC results. Bar graphs show the mean ± SEM of six biological replicates from two independent experiments. Asterisks show significance by unpaired $t$ test with Welch's correction (*$p = 0.0494$, **$p = 0.0017$)

CRY2 7m) or the tail residues (CRY1 C2T, CRY2 C1T) were swapped (Fig. 8a). All four constructs generated rescues with intermittent periods (CRY1 7m: 23.47 h, CRY1 C2T: 24.02 h, CRY2 C1T: 24.46 h, CRY2 7m: 24.13 h) between WT CRY1 (26.61 h) and CRY2 (22.89 h) (Fig. 8b), suggesting that neither the tail nor pocket is sufficient to fully recapitulate the periodicity phenotype of either WT CRY. However, chimeras in which both the pocket and tail were exchanged (Fig. 8c) were able to rescue rhythms with period and repression characteristics very similar to WT rescues (Fig. 8d, e). CRY1 7m C2T, which has both CRY2's pocket and tail, rescued rhythms with a significantly shorter period (21.78 h) than WT CRY2 (22.89 h) (Fig. 8e). In the case of CRY2 7m C1T, which has both CRY1's pocket and tail, the period (26.20 h) was indistinguishable from WT CRY1 (26.61 h) (Fig. 8e). Ultimately, these data suggest that both the tail and the pocket contribute to the periodicity of CRYs and both are required to fully capture native period characteristics.

## Discussion

CRY's evolution into a role as a direct repressor of CLOCK-mediated and BMAL1-mediated transcriptional activation suggests that specific structural changes have resulted in critical functional advantages. Much focus has been placed on the FAD-binding pocket of this family and significant work has emphasized the role of FAD in the function of plant CRYs and type I animal CRYs. Moreover, structural work from the past few years highlights the extent to which this pocket has been repurposed for protein–protein interactions with FBXL3 and PER2[20,23,24]. Here we describe how a hotspot of evolutionary changes in the CPF lineage, the secondary pocket, has also been repurposed for a crucial protein–protein interaction with the CLOCK/BMAL1 heterodimer.

CRY's C-terminal α-helix and tail have been shown to form a functionally important complex with BMAL1's TAD with a ~1–10 μM affinity[17,18,51]. Moreover, a recent report suggests that a common human mutation, which results in the deletion of 23 residues in CRY1's tail, results in stronger binding between CRY1 and the CLOCK/BMAL1 heterodimer[22]. Additionally, mutations on CLOCK's HI loop cause a complete abrogation of binding between CLOCK and CRY[18,52,53] and a recent report suggests that CLOCK's PAS-B domain, which contains the HI loop, interacts directly with the secondary pocket[19]. Taken together

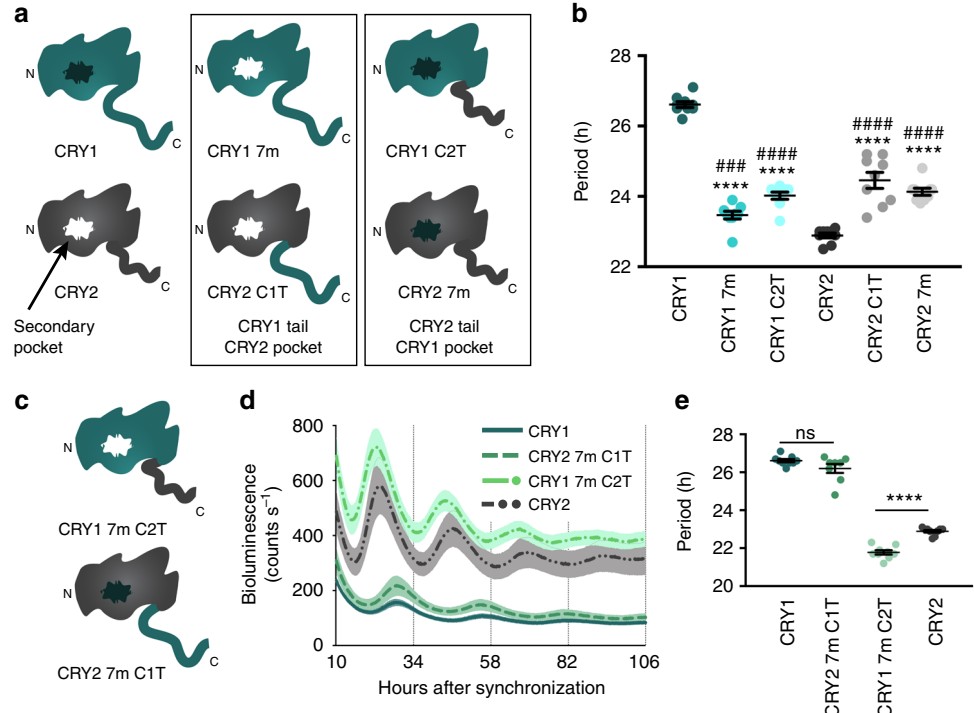

**Fig. 8** Both the CRY1 pocket and tail are required to recapitulate the long period length of *Cry1* rescues. **a** Models of chimeric CRYs used in the experiments in this figure. CRY1-like and CRY2-like pockets are shown in black and white, respectively. The first column shows WT CRY1 and CRY2; the second column shows CRYs with CRY1 tails and CRY2-like pockets; the final column shows CRYs with CRY2 tails and CRY1-like pockets. **b** Shows a period plot from experiments in which either the tail or pocket of CRY1 and CRY2 was exchanged for its paralog. Data are shown here as mean ± SEM ($n = 9$ plates per condition from three independent experiments. Asterisks show significance by unpaired *t* test with Welch's correction compared to WT CRY1 (****$p < 0.0001$). Hashes denote significance by unpaired *t* test with Welch's correction compared to WT CRY2 (###$p = 0.0006$, ####$p < 0.0001$). **c** Models depicting CRY chimeras in which both the pocket and tail have been exchanged for paralogous residues. **d** Rescue assays performed with WT CRY1, CRY2 7m C1T (CRY2 7m with CRY1 tail (res 498–606)), CRY1 7m C2T (CRY1 7m with CRY2 tail (res 516–593)), and WT CRY2 shown as mean ± SEM ($n = 9$ plates per condition from three independent experiments. **e** Period plot of the data shown in **d**. Mean ± SEM indicated by bars. Asterisks show significance by unpaired *t* test with Welch's correction (****$p < 0.0001$)

with our results, these data suggest a model in which an interaction between CLOCK's HI loop and CRY at the secondary pocket brings CRY into proximity with the BMAL1 TAD to create a sustained repressive complex. Tuning the affinity between CRY's CC helix and the BMAL1 TAD was shown to have a substantial effect on the period of the oscillation[18]. Similarly, we found that mutations in the secondary pocket that strengthen the interaction with CLOCK and BMAL1 lengthen the period of the oscillation and mutations that weaken the interaction predominantly shorten the period. Critically, the effects on period were independent of CRY's degradation rate. Historically, spontaneous and induced mutations in fly[54–57] and mammalian[10–12,14,15,48,58–61] genomes have helped to identify clock components that affect periodicity primarily by modulating the rate of decay of repressors. Our work demonstrates that the affinity of the core clock proteins for one another plays an important orthogonal role in determining periodicity and is in agreement with two other recent reports with similar findings[22,49]. Additionally, it is not yet clear whether modulating the degradation rate of CRY would rescue interactions between mutant CRYs and other core clock binding partners. One might expect that stabilizing CRYs with slightly weakened interactions would rescue periodicity by restoring the equilibrium of activator/repressor complexes. However, this idea has yet to be tested.

The structural underpinning of striking phenotypic differences in *Cry1*$^{-/-}$ and *Cry2*$^{-/-}$ mice has long been mysterious. At a behavioral level, deletion of one or the other leads to a

dramatically different output, but molecular analysis suggests a similar functional role. Amidst high sequence identity and similarity between the two proteins[62], no obvious structural differences present themselves as possible drivers of functional dissimilitude beyond the divergent tails. The data presented here strongly suggest that the functionality of these proteins stems in large part from the accumulation of subtle structural differences at the secondary pocket, which reduce the strength of the interaction between CRY2 and the CLOCK/BMAL1 heterodimer. Strikingly, co-expression of PER2 markedly improved the strength of the interaction between both CRY2 and the CRY1 7m mutant and the CLOCK/BMAL1 heterodimer. In concert with previous work demonstrating concurrent temporal occupancy for CRY2, PER1, and PER2 in chromatin association[37], our data suggest that CRY2 requires either PER1 or PER2 to form a stable repressive complex with CLOCK and BMAL1, restricting its repressive phase to the phase of PER expression. CRY1, due to its stronger physical interaction with CLOCK/BMAL1, is perhaps capable of forming a stable complex without a PER (Supplementary Fig. 9). How PER proteins mediate this stabilizing interaction is still unknown and a question for further exploration. In principle, this idea is similar to an idea from studies of circadian rhythms in *Neurospora crassa*, which posited that overall protein levels of the repressor FRQ do not determine period, but the levels of functional FRQ were highly determinant of period[63]. Likewise, if CRY2 is unable to bind CLOCK and BMAL1 without PER, then in the absence of PER, CLOCK, and BMAL1 are functionally blind to CRY2's presence. Additionally,

it is important to note that binding between CRY and PER may not always lead to repressive complex formation. Several studies have suggested that PER might also be involved in dissociation of the repressive complex, potentially by competing with BMAL1 for a binding interface on CRY[64–67]. Ultimately, the data shown here and elsewhere suggests a complicated and potentially heterogeneous role for PER in the molecular mechanism of the clock and clearly much work is needed to clarify PER's role.

One potential criticism of this model is that it discounts the dose–response experiments in Supplementary Fig. 7, which suggest that higher concentrations of CRY1 and CRY2 lead to shorter periods contradictory to the idea that a higher concentration of CRY1 would maintain a repressive state longer and delay the start of a new cycle of transcription, while higher levels of CRY2 would be irrelevant as PER2 is degraded. However, it is difficult to interpret the results of the dose–response experiments due to the fact that CRY will inhibit its own transcription. As a result, there is likely a floor and a ceiling for CRY expression in this system and increasing amounts of DNA merely change the dynamics of CRY expression, altering the rate of accumulation rather than the ceiling itself.

Ukai-Tadenuma et al.[46] demonstrated that a *ROR*-binding element (RRE) in an intron of the *mCry1* gene body is necessary for the delayed repressive phase of CRY1. A recent report by Edwards et al.[68], in contrast, showed that a small fragment of the *mCry1* promoter not including the RRE was sufficient to drive rhythmic expression of either CRY1 or CRY2 in brain slices containing the suprachiasmatic nucleus (SCN), a brain region that functions as a master oscillator. Moreover, they found that *Cry1* and *Cry2* were individually sufficient for rescue in *Cry1*−/−/*Cry2*−/− SCN and drove rhythmic output characteristic of *Cry2*−/− (long periods) or *Cry1*−/− (short periods) animals, respectively. Our results, in concert with these two reports, suggest that more work is necessary to understand the contributions of transcriptional regulation, protein–protein interaction dynamics, degradation, and intercellular coupling to periodicity in CRY1-driven and CRY2-driven rhythms. However, it is likely that each of these characteristics of the clock function as nodes for regulation of periodicity.

Our rescue data demonstrate that divergence at the secondary pocket drives a great deal of the repressive strength and periodicity differences between the two CRYs. However, it is also clear from rescues with CRY2 7m that this divergent structural feature cannot fully explain the differences in periodicity. The CRY1 tail and secondary pocket are together necessary and sufficient to fully convert CRY2's rescue profile to a CRY1-like profile. We have developed a mechanistic understanding of the secondary pocket's contribution to periodicity, but the mechanism underlying the tail's contribution is still unknown. It is, however, interesting to view the results of the chimera rescues in an evolutionary context. Exchanging either the tail or secondary pocket between the two CRYs invariably resulted in near 24 h rhythms, suggesting that in organisms with a single repressive CRY, such as the honey bee, one or the other of these features, but not both, is likely to remain intact. Given the low level of conservation in C-terminal tails, it is tempting to speculate that the delay characteristic of the CRY1 tail is in fact a gain-of-function mutation specific to organisms with two repressive CRYs. The nature of this function is a fertile area for future research.

Finally, the secondary pocket presents a potent opportunity for drug design. The data presented here suggest that antagonizing the interaction of CLOCK and BMAL1 at the secondary pocket has the potential to speed up the clock. Thus, small molecules designed to dock at this cavity could accelerate the clock, potentially functioning as a jetlag drug for west-to-east travelers. Advances in computational approaches to drug discovery require targeted selection of protein structural features based on a mechanistic understanding of the underlying biology. The data provided here form the basis for such further inquiry.

## Methods

**Resources**. Primers used in this study are detailed in Supplementary Table 2. Key resources used in this study are shown in Supplementary Table 3 along with manufacturer details where relevant.

**Experimental model and subject details**. *Cry1*−/−/*Cry2*−/− mouse embryonic fibroblasts were generated by Andrew Liu and Hiroki Ueda[46] and were gifted to us. HEK-293A cells were purchased from Thermo Fisher Scientific. Both lines were grown in Dulbecco's modified Eagle's medium (DMEM) containing 10% fetal bovine serum (FBS) and 1× Pen/Strep antibiotics at 37 °C under 5% CO₂. Lumi-Cycle recording medium was prepared from powdered DMEM without phenol red containing 4.5 g/L glucose and supplemented with 10 mM HEPES, pH 7.20, 100 μM luciferin, 1 mM sodium pyruvate, 0.035% sodium bicarbonate, 2% FBS, 1× Pen/Strep antibiotics, and 2 mM L-glutamine. LumiCycle recordings were performed at 37 °C.

**Statistical coupling analysis**. Approximately 10 000 CRY and PHL sequences were collected from NCBI on March 30, 2016 using the full-length mCRY1 protein sequence (GenBank Accession ID: AAH85499.1) as a search sequence for a PSI-BLAST. PSI-BLAST parameters were adjusted from default to include an Expect Threshold of 0.01, a PSI-BLAST Threshold of 0.005, and a minimum sequence identity of 20%. PSI-BLAST was performed iteratively for two rounds before the sequences were downloaded. The sequences were initially filtered by size using a custom python script to remove sequences smaller than 400 or larger than 800 residues. Remaining sequences were subjected to an initial alignment using the alignment tool MUSCLE[69] locally for two iterations. Extraneous header information was removed using a custom python script before performing a secondary alignment using Promals3D with six additional structure sequences (cyclobutane pyrimidine dimer (CPD) PHL from *Escherichia coli*, CPD PHL from *Anacystis nidulans*, 6–4 PHL from *Arabidopsis thaliana*, CRY from *Drosophila melanogaster*, and CRY1 and CRY2 from *Mus musculus*[24,26,51,70–73]. Using a custom python script, GI numbers were collected from each header in a separate file for use in the annotation step of the SCA. SCA calculations were performed using the pySCA toolbox as described[44]. The annotation step was performed through NCBI using the GI numbers on April 17, 2016. The initial alignment of 9719 sequences consisted of 4344 independent positions. The alignment underwent preprocessing during which highly gapped positions were removed. After removing highly gapped *positions*, remaining highly gapped *sequences* were removed using a set cutoff. Sequences with too great of a fractional identity to the reference sequence were also removed. Finally, each sequence was weighted based on the number of sequences with an identity above 80% to the given sequence. This final weighting step allows an effective number of sequences (M′) to be computed based on the remaining number of actual sequences (M). After all of the preprocessing steps, the final alignment contained 5385 sequences representing 2447 effective sequences composed of 459 positions. Critically, an alignment should be large and diverse enough to give reasonable estimates of amino acid frequencies. Two thousand four hundred and forty-seven effective sequences is well above the suggested minimum of 100 effective sequences needed to obtain a reasonable estimate[44]. During these processing steps, default parameters were used and residues were mapped to the CRY2 structure (PDB: 4I6E)[24]. Following initial calculation steps, the workflow was performed in an ipython notebook.

Here we provide a brief overview of the SCA calculations; the complete mathematical formalism and methods as used here are more completely presented in ref [44]. Conservation values for individual amino acid positions were calculated as a Kullback–Leibler relative entropy ($D_i$), a measure that quantifies the divergence of amino acid frequencies at position $i$ from the distribution of amino acid frequencies expected randomly. Values of $D_i$ near zero indicate that amino acid frequencies at that site approach random expectation, while values of more than 3 indicate highly conserved positions (>~83% residue identity). Co-evolution between all pairwise combinations of amino acids over all $L$ positions was computed as an $L \times L \times 20 \times 20$ conservation-weighted co-variance matrix:

$$\widetilde{C_{ij}^{ab}} = \frac{\partial D_i^a}{\partial f_i^a} \frac{\partial D_j^b}{\partial f_j^b} \left( f_{ij}^{ab} - f_i^a f_j^b \right), \qquad (1)$$

where $f_i^a$ indicates the frequency of amino acid $a$ at site $i$ (see also ref. [44]). The basic idea is that positions experiencing a joint functional constraint will co-vary in an alignment; weighting the co-variance by the degree of conservation $(\partial D_i^a/\partial f_i^a)$ serves to emphasize couplings between functionally relevant positions and helps reduce phylogenetic noise. This matrix was then dimension reduced to produce the

final SCA matrix:

$$\widetilde{C_{ij}} = \sqrt{\sum_{a,b} \left(\widetilde{C_{ij}^{ab}}\right)^2}, \qquad (2)$$

In order to identify correlated groups of positions within the protein, we analyzed the SCA matrix $\widetilde{C_{ij}}$ using spectral decomposition. We defined a single sector along the top eigenmode (EV1 > 0.03) as shown in Figs. 1 and 2 and Supplementary Table 1. The top two eigenvectors of the positional correlation matrix were then used to project the correlations between sequences to arrive at Fig. 2b, c.

**Site-directed mutagenesis**. Mutagenesis was performed using both the Quik-Change II XL kit and, with a few modifications, the Q5 Site-Directed Mutagenesis kit. Manufacturer's instructions were followed for QuikChange mutagenesis. For Q5 Site-Directed Mutagenesis, primers were designed using the NEBaseChanger tool (http://nebasechanger.neb.com/). Template DNA was mixed with 2× Q5 polymerase chain reaction (PCR) Master Mix and primers before being subjected to PCR. Following PCR, 1 μL of each PCR product was combined with 2× Quick Ligation buffer and ligase, along with 1 μL of T4 polynucleotide kinase, and 1 μL of *Dpn*I in a total of 10 μL. The ligation reaction was incubated at room temperature for 5 min and 5 μL of product were used to transform DH5α-competent cells. Colonies were selected for culturing and miniprep, and mutations were verified by Sanger sequencing.

**Gibson assembly cloning**. CRY tail chimera rescue vectors were generated by Gibson assembly cloning using NEBuilder HiFi DNA Assembly kit. Briefly, the rescue vector containing *Cry1*, *Cry2*, *Cry1 7m*, or *Cry2 7m* was linearized by PCR with primers, which also removed the coding sequence for residues 498–606 of *Cry1* and *Cry1 7m* or residues 516–593 of *Cry2* and *Cry2 7m*. The C-terminal Myc tag was left intact. The coding sequences for the tail regions of *Cry1* (residues 498–606) and *Cry2* (residues 516–593) were amplified by PCR using primers that would generate overlaps between the amplified tail product and the linearized target vectors. PCR products were purified using the QIAquick PCR Purification kit and combined with inserts and linearized target vectors in a 2:1 molar ratio. These combined products were then treated with NEBuilder HiFi DNA Assembly master mix containing an exonuclease, DNA polymerase, and DNA ligase to induce assembly of the final vector. This solution was incubated at 50 °C for 15 min and the products were used to transform DH5α competent cells. Colonies were selected for culturing and miniprep, and insertions were verified by Sanger sequencing.

**Real-time bioluminescence rescue assays**. A total of $4 \times 10^5$ *Cry1$^{-/-}$/Cry2$^{-/-}$* mouse embryonic fibroblasts were plated in 35 mm tissue culture dishes and transfected the same day with 4 μg of a luciferase reporter (pGL3-P(Per2)-Luc) and 150 ng of a *Cryptochrome* rescue vector (pMU2-P(Cry1)-(intron336)-Cry-Myc, modified with a C-terminal MYC tag)[46] using FuGENE 6 according to the manufacturer's instructions. At 72 h after transfection, the cells were synchronized by exchanging growth medium for growth medium supplemented with 0.1 μM dexamethasone and returned to the incubator for 2 h. The medium was then replaced by LumiCycle recording medium and the plates were sealed with vacuum grease and cover glass and transferred to the LumiCycle. Bioluminescence monitoring was performed using a LumiCycle to record from each dish continuously for 70 s every 10 min using a photomultiplier tube at 37 °C. Rescue results were processed using the LumiCycle Analysis software package. The first 10 h of recording were discarded and period, amplitude, phase, and damping rate were calculated using a damped sine wave based on a running average fit for each plate of cells. Rescues were considered arrhythmic if the goodness of fit for the damped sine wave was <80%. Subsequent data were corrected for background noise in each photomultiplier tube channel, which was measured by monitoring an untransfected plate of cells for 24 h and averaging the signal. All CRY1 and CRY2 mutants were run with WT CRY1 as an internal control. In some cases, multiple experiments were combined in the data shown if there were positive internal controls for each dataset. Rescues were excluded only in cases where the transfection did not work, which was obvious based on a very weak or non-existent luminescence signal and the lack of induction of luminescence shortly after synchronization. Appropriate replicate size was determined empirically from extensive use of these assays in previous work[20].

**Vector construction**. Full-length *mCry2* cDNA was cloned into the pMU2-P(Cry1)-(intron336)-Myc rescue vector backbone using megaprimer mutagenesis. Luciferase-tagged constructs were also generated using megaprimer mutagenesis. Briefly, the relevant insert was copied from a source vector by PCR (using PfuUltra II Fusion HS DNA polymerase with primers containing roughly 30 bp of overlap in either direction with the insertion site in the target vector and 30 bp of overlap with the insert). Following primary PCR, PCR products were purified using the QIA-quick kit, and 400 ng of purified product used as a megaprimer for a secondary PCR with the target vector serving as a template. PCR products were treated with

*Dpn*I for several hours, and then DH5α-competent cells were transformed with 5 μL. Colonies were selected for culturing and miniprep and insertions were verified by Sanger sequencing. *Photinus pyralis Luciferase* was fused directly to the C terminus of *Cry1* and *Cry2* in the pCMV-Tag3C-Myc vector. Full-length *Cry2* was inserted at the exact location in pMU2-P(Cry1)-(intron336)-Myc where *Cry1* was deleted.

**Immunoprecipitation**. HEK-293A cells were seeded in 6-well tissue culture plates at a density of $4 \times 10^5$ cells per well. Cells were transfected the same day with FuGENE 6. The following constructs were used for all transfections: p3X-FLAG-CMV-10-*mBmal1*, p3X-FLAG-CMV-10-*mClock*, pCMV-Myc-*Cry1*[21], pCMV-Myc-*Cry2*[21], and pcDNA3.1-*mPer2*-V5[74]. For IPs involving all four clock components, cells were transfected with 45 ng of *mBmal1*, 1.5 μg of *mClock*, 600 ng of *mPer2*, and 150 ng of either *Cry1*, *Cry2*, or mutants thereof, in a total of 2.295 μg of DNA. Empty vector (pcDNA3.1-A) was used to even out DNA cocktails. For IPs of only BMAL1, CLOCK, and CRY, cells were transfected with 45 ng of *mBmal1*, 1.5 μg of *mClock*, and 150 ng of *Cry* plasmid in a total of 1.695 μg of DNA. Finally, for IPs of PER and CRY alone, cells were transfected with 600 ng of *mPer2* or 300 ng of *Cry* in a total of 900 ng of DNA. After 48 h, cells were lysed in 200 μL of TGED buffer (50 mM Tris-HCl, pH 7.5, 100 mM NaCl, 5% glycerol, 0.5 mM dithiothreitol, 0.5% Triton X-100, protease inhibitor (1:50)) for 30 min before a 10 min centrifugation to remove cellular detritus. Ten percent of the supernatant solution was saved for use as input and the remainder was incubated for 3 h with 40 μL of anti-MYC-conjugated beads to immobilize MYC-CRY. Beads were washed with 1 ml of TGED buffer twice. Protein was released from beads by boiling in 50 μL of sodium dodecyl sulfate (SDS) sample buffer (26.3 mM Tris-HCl, pH 6.8, 4.2% glycerol, 0.84% SDS, 10.5% β-mercaptoethanol, 0.21 mg/mL bromophenol blue) and analyzed by immunoblot using anti-MYC, anti-V5, and anti-FLAG-HRP (horse radish peroxidase) for CRY, PER2, and CLOCK/BMAL1, respectively. Catalog numbers for the antibodies can be found in Supplementary Table 3. For immunoblots of all four proteins: anti-MYC was used at a concentration of 1:4000 on input blots and between 1:25 000 and 1:80 000 on IPs; anti-FLAG-HRP was used at a concentration of 1:200 000 on input blots and between 1:3000 and 1:100 000 on IPs; anti-V5 was used at a concentration of 1:2 000 000 on input blots and 1:10 000 000 on IPs. For immunoblots of CLOCK, BMAL1, and CRY: anti-MYC was used at a concentration between 1:3000 and 1:5000 on input blots and between 1:20 000 and 1:50 000 on IPs; anti-FLAG-HRP was used at a concentration between 1:10 000 and 1:50 000 on input blots and between 1:1000 and 1:5000 on IPs. For immunoblots of CRY and PER2: anti-MYC was used at a concentration of 1:5000 on input blots and 1:50 000 on IPs; anti-V5 was used at a concentration between 1:5000 and 1:25 000 on input blots and between 1:50 000 and 1:200 000 on IPs. The anti-mouse immunoglobulin G, HRP-linked secondary was always used at a concentration of 1:10 000. Blots were imaged on radiography film with Clarity ECL Substrate. Appropriate replicate size was determined empirically from extensive use of these assays in previous work[20].

**Real-time bioluminescence degradation assays**. HEK-293A cells were plated in 35 mm tissue culture dishes at a density of $4 \times 10^5$ cells per well. Cells were transfected (FuGENE 6) same day with 500 ng of a *Cry-Luc* fusion construct (pCMV-Myc-*Cry1-Luc* or pCMV-Myc-*Cry1-Luc*) and 200 ng of an EGFP construct (pEGFP) as a transfection control. At 48 h after transfection, the medium was exchanged for LumiCycle recording medium supplemented with cycloheximide (100 μg/ml). Plates were transferred to the LumiCycle and bioluminescence was monitored from each dish continuously for ~70 s every 10 min using a photomultiplier tube at 37 °C. Data were corrected for background noise from the photomultiplier tube as described above. The first 10 h of recording were normalized to the first data point and half-life was determined by nonlinear, one-phase exponential decay analysis.

**Alignments**. CRY alignments for Fig. 4 and Supplementary Fig. 3 were performed with CLC Main Workbench 7 using default settings for the slow alignment mode. Sequences were accessed and downloaded from NCBI.

**Bimolecular fluorescence complementation**. Full-length Venus (Ex515/Em528) and Cerulean (Ex433/Em475) have been described previously[75]. PCR-amplified *Cry1* and *Cry2* cDNAs were inserted into the *NheI* and *AgeI* sites upstream of the coding region of CerN and VenN. Full-length *Clock* cDNA was amplified by PCR with an N-terminal *SacII* site and a C-terminal blunt polymerization end and ligated into *SacII* and *SmaI* sites in CerC. Residues 2–88 and 396–855 of *Clock* were deleted from the construct by site-directed mutagenesis. Truncated Venus (VenN (1–155 amino acids)) and Cerulean (CerN (1–155 amino acids), CerC (156–239 amino acids)) fragments were previously generated in the pEGFP-C1 (Clontech) backbone by site-directed mutagenesis[15].

For BiFC competition experiments, 25 ng H2B-mRFP1[76] was mixed with 200 ng CerC-Clock (89–395), 10 ng p3X-Flag-Bmal1, 200 ng of Cry1-VenN or Cry1-CerN, and 200 ng of Cry2-CerN or Cry2-VenN in a total of 735 ng of DNA. A total of $8 \times 10^4$ HEK-293A cells were plated in each well of a 24-well black Visiplate. On the same day, cells were transfected with FuGENE 6. Plates were washed with phosphate-buffered saline (PBS) once 48–60 h after transfection, fixed with 4%

paraformaldehyde in PBS for 15 min, and then immersed in PBS. A control well for background fluorescence was transfected with non-complementary BiFC vectors (200 ng Cry1-VenN, 200 ng Cry2-CerN, 25 ng H2B-mRFP1). Controls for normalization of the competition experiments were transfected with single complementation pairs (25 ng H2B-RFP1, 200 ng CerC-Clock (89–395), 10 ng p3X-Flag-Bmal1, and 200 ng of Cry2-CerN or Cry2-VenN in a total of 735 ng of DNA).

Fluorescence images were acquired on a Deltavision Personal DV Imaging System equipped with an inverted 10× (for quantification) or 20× (for publication images) 0.45NA UPLFL objective and a Microtiter stage on an Olympus IX71 microscope. Four locations in each well were selected and autofocused with red fluorescent protein 1 (RFP1) fluorescence. Single layer images for quantification or 20-layer Z stacks with 1 mm steps for deconvolution and presentation were scanned in the channel sequence of RFP (Ex575/25; Em632/60), YFP (Ex513/17; Em559/34), and CFP (Ex438/24; Em465/30) filter sets with the Z step first. Image stacks were deconvoluted with the Softworx deconvolution module and maximal intensity Z projections of layers were built.

For quantification of BiFC competition results, original image files were imported to and organized by ImageJ (NIH), and exported image sequences were pushed through a custom pipeline run by Cellprofiler. Briefly, the nuclei were first recognized based on RFP fluorescence, and then inverse-masked with strong aggregates in the Venus and Cerulean channels. Mean Venus and Cerulean intensity values of each masked nucleus were measured. About 40–120 cells were identified and measured in each image, and the average values of cells in one image were obtained for further normalization and statistical analysis. Background intensity values from Venus and Cerulean channels were subtracted from competition experiment intensity values. Background subtracted competition values were then normalized to background subtracted intensity values from single complementation wells. Four images from each well were averaged and the experiments were replicated three times.

**Quantification and statistical analysis**. Statistical parameters are reported in the figure Legends and indicated in the figures where appropriate. Unpaired $t$ tests with Welch's correction were performed by GraphPad Prism software. Welch's $t$ test was chosen to compare the period means and half-life means because we do not know whether the various mutant populations have equal variances to the WT protein. Protein half-lives were determined by normalizing each luminescence value of a given sample to the initial luminescence reading, and then fitting a one-phase decay curve to the resulting data in GraphPad Prism.

**Code availability**. Custom scripts used in processing the SCA data, the alignment used for the SCA, the annotated SCA notebook, and the SCA database file are all available upon request. The custom data pipeline for CellProfiler is available upon request.

**Data availability**. Other data are available from the corresponding author upon reasonable request.

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

## Acknowledgements

We thank Andrew C. Liu and Hiroki Ueda for the generous gift of the $Cry1^{-/-}/Cry2^{-/-}$ mouse embryonic fibroblasts and the *Cry1* rescue vector. Research was supported by NIH grants (F31 NS089241 and T32 HL007909 to C.R.; R01 GM090247, R01 GM111387 and R01 GM112991 to C.B.G.; K.A.R. was supported by the Green Center for Systems Biology. J.S.T. is an Investigator in the Howard Hughes Medical Institute. R.R. acknowledges support from the NIH (RO1GM123456), the Robert A. Welch Foundation (I-1366), and the Lyda Hill Endowment for Systems Biology.

## Author contributions

Conceptualization, C.R., J.S.T., and C.B.G.; Methodology, C.R., J.S.T., and C.B.G.; Software, K.A.R. and R.R.; Formal Analysis, C.R. and K.A.R.; Investigation, C.R., I.L., and P.G.; Resources, P.G., Y.S., J.S.T., and C.B.G.; Writing—Original Draft, C.R. and C.B.G.; Writing—Review and Editing, C.R., K.A.R., P.G., R.R., J.S.T., and C.B.G.; Visualization, C.R., K.A.R., and R.R.; Supervision, J.S.T. and C.B.G.; Funding Acquisition, C.R., K.A.R., R.R., J.S.T, and C.B.G.

## Additional information

**Competing interests:** The authors declare no competing interests.

