## [Peer Review File(PDF 454 kb) · Nature Communications]

Reviewers' comments:

Reviewer #1 (Remarks to the Author):

Cryptochrome is the key molecule that regulates the period of circadian clocks, and thus its circadian-functional domain has been extensively studied. Many of the previous studies focused on the primary co-factor binding domain in CRYs, but a recent study by Michael et al pointed out the importance of the secondary co-factor pocket. In this study from Green's lab applied well-informed Statistical Coupling Analysis (SCA) to reveal the possible functional residues co-evolved among cryptochrome family proteins and suggested that secondary co-factor pocket may function as an allosteric regulation site not only in PHL but also in CRY. A comprehensive mutagenesis of residues around the secondary pocket in mammalian CRY confirmed that this pocket indeed regulates the circadian period. Interestingly, the period modulation can be attributed to the changes in the affinity to BMAL1/CLOCK rather than the well-established correlation between period and CRY1's protein stability. The authors further pointed out that part of the difference between CRY1 and CRY2 can be attributed to the altered residues around this pocket. Although most of the concept proposed in this study has already been reported separately (i.e. i: uncoupling of CRY stability and period modulation [PMID: 28388406, 28017587], ii: role of PER protein in the CRY-BMAL1/CLOCK interaction [PMID: 27688755, 25228643, 21613214, 24794436], iii: role of secondary pocket and CRY-BMAL1/CLOCK interaction in the regulation of circadian period [PMID: 28143926], iv: identification of residues differentiate CRY1 and CRY2 [PMID: 22692217], v: correlation between period and CRY-BMAL1/CLOCK interaction [PMID: 25961797, 28388406]), this study has a potential to connect them in the view of secondary cofactor pocket and protein-protein interaction. I only have following minor comments, which may further improve the manuscript. Overall, the study will contribute to the understanding of the mechanism by which CRYs regulate the circadian period.

Minor comments:

- 1) Is it possible to dissect the feature of mammalian Cryptochrome by the SCA analysis (figure 2)? How mammalian CRY differentiates from the other CRY/PHL to function as a circadian transcriptional repressor shall be interesting for the circadian researchers. Analysis on figure 2 merged all metazoan and figure 3 merged all CRY, thus it might not suit to characterize the mammalian CRY.
- 2) It is important to quantitatively/statistically check the correlation between the period and the CRY-BMAL1/CLOCK affinity. A presentation like a fig. 6H may be suitable for this purpose.
- 3) The authors are requested to provide a rationale why they chose CLOCK not BMAL1 (or CLOCK-BMAL1) to analyze the CRY1-interaction in the BiFC assay.
- 4) As for the CRY2 7m C1T and the CRY1 7m C2T, are there published studies suggesting the role of CRY c-terminal domain in the interaction with BMAL1-CLOCK? They should be mentioned in the discussion section.
- 5) The authors may want to discuss the model of PER proposed in figure S9 based on the context-dependent action of PER. Serial studies from Sancer's group and others have proposed that PER can repress and de-repress the BMAL1-CLOCK through CRY1 (PMID: 25228643, 27688755, 21613214, 24794436). This model is conceptually similar to the model proposed in this study, but figure S9 premises the interaction of PER-CRY always leads to a transcription-repressive complex—which may not be always true.
- 6) Line 703. It would be fair to clearly mention that half-life independent period control by CRY is highlighted in recent studies (PMID: 28388406, 28017587), in particular, one of them (28388406)

connects the half-life independent mechanism to the interaction between CRY1 and BMAL1/CLOCK.

7) An "introduction" part from line 125 to line 181 is a bit verbose for the result section. It can be summarized and/or part of this section can be moved to the introduction.

Reviewer #2 (Remarks to the Author):

The manuscript by Rosenzweig et al provides an in depth per residue analysis of the role of the "secondary pocket" of the mammalian cryptochromes CRY1 and CRY2. This pocket binds an antenna chromophore (e.g. MTHF, 8-HDF) in DNA repairing photolyases. The authors show that the secondary pocket, in addition to the tail, defines distinct functions of the CRY1 and CRY2 homologues in the mammalian circadian clock.

The authors use Statistical Coupling Analyses (SCA) to define a (potentially allosteric) network of coevolving residues that includes the FAD binding pocket as well as the secondary pocket.

The authors use Co-IP- and rescue experiments to validate the role of residues within and surrounding the secondary pocket in CLOCK/BMAL1- and PER binding as well as in maintaining circadian oscillations.

Comparison with photoreceptor-type insect cryptochromes suggests that mammalian CRYs have developed a more open secondary pocket for CRY-CLOCK/BMAL1 interactions. The manuscript provides experimental evidence for this concept by mutation of secondary pocket residues of CRY1 to insect CRY residues (G106W, F41S, P39G).

Furthermore, the authors identify seven residues at the secondary pocket that together with the tail define different functions of CRY1 and CRY2. They also show that a CRY1-like secondary pocket strengthens CLOCK/BMAL1 interactions in absence of PER2, while a CRY2-like pocket defines the need of PER2 for stable BMAL1/CLOCK interactions. Further, the authors show with BIFC that CRY1 interacts more strongly with a CLOCK (PAS-A/B) fragment than CRY2.

General evaluation:

The new findings described in this manuscript are very important to understand the mechanistic basis of CRY1 and CRY2 function in the circadian clock, specifically the role of the secondary pocket. Moreover, the manuscript provides essential information to explore the secondary pocket as a potential drug target to speed up the clock, e.g. to overcome jetlag. The presented work significantly advances our general mechanistic understanding of circadian clock operation, especially the role of protein-protein interactions in circadian regulation. The manuscript is of broad interest, including circadian research and the mechanistic understanding of cryptochrome/photolyase family members. The techniques are well described.

Before publication, the following issues should be addressed:

General issue: The authors should be more careful using the term "allostic/allostery" (see detailed examples below). Also, the previously published data on the second pocket (Nangle et al, 2014; Michael et al, 2017, Czarna et al, 2013) should be presented more clearly to make it clearer what was already known before and what is new in this manuscript (see detailed below).

Some detailed issues:

Abstract, Line 38: "weaker interactions – of CRY2 ? - with CLOCK/BMAL1" ? Please insert "of CRY2", if this is what the author's meant to say. As it is, this sentence is not clear.

Line 66 introduction "nevertheless it remains unclear which CRY interfaces lead to specific protein-protein interactions with CLOCK and BMAL1." The authors place this statement in the introduction without acknowledging that a previous publication by Michael et al, PNAS 2017 already reported the

importance of the CRY secondary pocket for Clock-PAS-B-HI loop interactions. Furthermore, several earlier publications e.g. by Chaves et al, 2006, Xu et al, 2013, Czarna et al, 2011 have already reported the role of the C-terminal coiled-coil helix of the CRY-PHR and the CRY-tail in transcriptional repression via the BMAL1-TAD.

It would be good if the authors would mention some of these earlier findings in the introduction, and not only in the discussion (Lines 686-691), to more clearly emphasize the new aspects of this manuscript.

Line 160, p.8: In Photolyases, the second cofactor (e.g. MTHF) transfers energy to the FAD cofactor via Förster Resonance Energy Transfer. I would not call this an "allosteric" modulation of the dynamics of the repair reaction, as the two chromophores and the DNA-lesion directly communicate with each other. Please remove "allosterically" in line 160.

Fig. 1C and Mat/meth p.38, line 831: please define "Di". What does $D_i = 1,2,3 \dots$ mean in terms of conservation? What does "weakly/moderately/more conserved" mean in Fig. 1C legend?

Fig. 1C legend: Please define "CPF" at first mention (it is defined later in the text)

Line 172: The dCRY structure presented in Zoltowski et al, Nature 2011 (Ref. 34) is not correct. The authors should cite Levy et al, Nature 2013 and Czarna et al, Cell 2013, which report correct dCRY structures.

Line 214: what do the authors mean by "the known allosteric mechanisms linking the two" (i.e. the FAD and secondary binding pocket)? I would not call energy and electron transfer in the repair reaction of photolyases an allosteric mechanism (see above).

Line 261: see comment on line 214.

Fig. 3A: it may be helpful to show the "lower helix" as ribbon in addition to sticks.

Line 306: The authors should also mention the published results on R109Q regarding its reduced repression activity in circadian rescue experiments (Nagle et al, Elife 2014). This provides earlier evidence for the role of the secondary pocket in transcriptional regulation (see also line 130). It would be easier for the reader if earlier reported results on R109Q (Nangle et al, 2014; Michael et al, 2017), which were presented in the context of CRY1/2 crystal structures, would be placed in one location in the new manuscript.

Line 337: looking at the alignments that the authors present I would not polarize the level of conservation between the upper and lower helix so much. There are exchanges in both motifs. So "largely conserved" vs. "highly divergent" is more contrasting than the alignments indicate.

Line 687: I would not call a 1 to 10 μM affinity a "high affinity". High affinity would rather imply a nM range affinity than a μM range affinity.

Line 692: Are there nM range KD values available for the "high affinity interaction between CLOCK's HI loop and CRY at the secondary pocket"? If not, I would not classify this interaction as "high affinity". If this interaction has not been quantified by determining KD values, it is dangerous to postulate, that the CLOCK-PAS-B-HI-loop/CRY-secondary pocket interaction recruits CRY primarily to the BMAL1/CLOCK complex and that the lower affinity (μM) BMAL1-TAD-CRY-tail interaction is secondary- unless the authors have other evidence for this model. Also "allostery" (line 694) would imply an indirect conformational effect of the CLOCK-CRY interaction on the BMAL1-TAD-CRY-tail interaction. Is there evidence for this? If not I would rank it as a proximity effect rather than an allosteric effect.

Fig S5: The difference between Figure S5C and Fig. S5D/E is not clear. Please explain in Fig. legend.

Reviewer #3 (Remarks to the Author):

Rosensweig et al. present a study that is important in many aspects: (i) it provides a detailed molecular understanding of the role of cryptochromes in the mammalian circadian clock; (ii) it sheds light on the evolution of cryptochromes with regard to their functional diversification.

Circadian clock regulation in mammals has still several mysteries, in particular with respect to cell-biological, but also with respect to structural aspects. Within the recent years, much has been learned about structural properties of cryptochromes also in complex with binding partners or small molecules regulating their stability. However, one puzzling aspect was largely unknown (at least from a structural point of view): Cry1 knockout leads to short periods, and Cry2 knockout to long periods (both in behaviour and cells). This study solves this puzzle by showing that the so-called secondary pocket in cryptochromes (i.e. residues therein) is divergent between CRY1 and CRY2 and modulates the binding strength to CLOCK thereby differentially inhibiting the transactivation activity of CLOCK/BMAL1.

In my opinion, this is a well-performed important study that eventually should be published. However, I have a couple concerns and suggestions the authors might want to consider.

1. SCA: (i) It has been known before that CLOCK binds to the secondary pocket of CRYs. 260 residues were identified to be sector residues – a very large fraction of the surface-exposed CRY residues. I don't really see a major benefit of the SCA for this particular study: the structures are known; residues of the secondary pocket could have been selected without SCA. At least, it would be helpful to see (in a table), how many residues of the secondary pocket are sector residues, how many and which of them have been mutated and tested and which ones resulted in an effect. I concede the evolutionary aspect is interesting, but may deserve a separate publication. In essence, do you really need SCA to study the importance of the secondary pocket? (ii) The presentation of SCA is lengthy (2 figures!) and sometimes difficult to understand for the non-expert reader – this should be improved.
2. Allosterity: (i) The functional sites of mammalian CRYs should be better explained in the introduction, e.g. what is the flavin pocket doing in mammals? Why is this the active site? (ii) It is unclear, how mutations at the secondary pocket influence the "active" site. It is obvious that they modulate binding affinity to CLOCK, but it is unclear whether affinity to BMAL1-TAD is allosterically regulated. Is it really allosteric regulation in mammalian CRYs or is this a "historic" term referring to pterin's role of photon capture and transmitting energy to flavin in the photolyase's active site?
3. The authors should better discuss or even might want to test the role of CRY stability in this context. E.g. can a mutant with very weak CLOCK binding properties (e.g. D38A) be rescued by Fbx13 knockdown? Would Fbx13 knockdown have a differential effect for Cry1 and Cry2 knockouts?
4. Overall, there seems to be a relation between rescue period, repression activity and binding to CLOCK. Sometimes, however, this does not apply (Figure 5B,C and Figure 3E,F). What are the other determinants? E.g. are K107 and E108 mutants more stable? In general, the authors use primarily three types of assays: (i) rescue, which provides period and repression; (ii) binding assays with IP and (iii) stability measurements. It is unclear, why these assays are not always used for all mutants tested to get a comprehensive picture.
5. The bimolecular fluorescence complementation assay is elegant and should be further exploited to test, whether PERs indeed strengthen the binding of CRY2 to CLOCK.
6. Figure 8: Does the tails modulate binding to CLOCK/BMAL1?

Response to Reviewers

We are extremely grateful to these three excellent reviewers for the many constructive comments and suggestions. Our response to each comment is shown below (bold text with yellow highlight). The changes to the manuscript are also marked by yellow highlighting for easy viewing.

Reviewer #1 (Remarks to the Author):

Cryptochrome is the key molecule that regulates the period of circadian clocks, and thus its circadian-functional domain has been extensively studied. Many of the previous studies focused on the primary co-factor binding domain in CRYs, but a recent study by Michael et al pointed out the importance of the secondary co-factor pocket. In this study from Green's lab applied well-informed Statistical Coupling Analysis (SCA) to reveal the possible functional residues co-evolved among cryptochrome family proteins and suggested that secondary co-factor pocket may function as an allosteric regulation site not only in PHL but also in CRY. A comprehensive mutagenesis of residues around the secondary pocket in mammalian CRY confirmed that this pocket indeed regulates the circadian period. Interestingly, the period modulation can be attributed to the changes in the affinity to BMAL1/CLOCK rather than the well-established correlation between period and CRY1's protein stability. The authors

further pointed out that part of the difference between CRY1 and CRY2 can be attributed to the altered residues around this pocket. Although most of the concept proposed in this study has already been reported separately (i.e. i: uncoupling of CRY stability and period modulation [PMID: 28388406, 28017587], ii: role of PER protein in the CRY-BMAL/CLOCK interaction [PMID: 27688755, 25228643, 21613214, 24794436], iii: role of secondary pocket and CRY-BMAL/CLOCK interaction in the regulation of circadian period [PMID: 28143926], iv: identification of residues differentiate CRY1 and CRY2 [PMID:

22692217], v: correlation between period and CRY-BMAL/CLOCK interaction [PMID: 25961797, 28388406]), this study has a potential to connect them in the view of secondary cofactor pocket and protein-protein interaction. I only have following minor comments, which may further improve the manuscript. Overall, the study will contribute to the understanding of the mechanism by which CRYs regulate the circadian period.

We thank the reviewer for these supportive comments.

Minor comments:

1) Is it possible to dissect the feature of mammalian Cryptochrome by the SCA analysis (figure 2)? How mammalian CRY differentiates from the other CRY/PHL to function as a circadian transcriptional repressor shall be interesting for the circadian researchers. Analysis on figure 2 merged all metazoan and figure 3 merged all CRY, thus it might not suit to characterize the mammalian CRY.

We agree with the reviewer that further analysis of mammalian-specific sequence changes would be highly interesting. However, the statistical coupling analysis (SCA) requires a large and diverse sampling of sequences in order to detect a significant signal. Following alignment processing to remove highly similar sequences (within 90% of mouse CRY2) we are left with only 18 mammalian sequences – a insufficient number for good statistics.

2) It is important to quantitatively/statistically check the correlation between the period and the CRY-BMAL1/CLOCK affinity. A presentation like a fig. 6H may be suitable for this purpose.

Our quantifications are based on western blots, which are an imperfect assessment of binding affinity. However, given the number of mutants we

have characterized and the difficulty of purifying CRY, CLOCK, and BMAL1 at a scale necessary to perform quantitative binding assays, immunoprecipitation was the only reasonable choice available for these experiments. We have prepared figures like fig. 6H to compare the ratio of CLOCK or BMAL1 to CRY and the periodicity observed in the rescue assay (presented below). The only appropriate comparison is for the mutations in which we performed IPs without PER2 coexpression, so we are limited to R51A, E382A, F257A, CRY1 7m, CRY2 7m, and WT CRY1 and CRY2.

We performed a linear regression on this data and found that the data were poorly fit ($R^2 = 0.1644$ and 0.3137 for CLOCK and BMAL1 respectively). However, it is clear that CRY2 7m is an outlier and causes substantial deviation in the overall goodness of fit of our linear regression. Removing this single data point substantially improves the fit ($R^2 = 0.7018$ and 0.7301 respectively). Ultimately, these data (together with the analysis of degradation rates) support the idea that periodicity is a complex trait that is imperfectly described by any one characteristic of the system. There are likely to be known and unknown factors that also influence periodicity, conversant with the parameters described herein. However, we feel that these data suggest that differences in affinity best describe the mechanism underlying the range of periodicities observed during our experimentation.

The ratio of CLOCK to CRY or BMAL1 to CRY (reported in supplementary figures 5E and 8C) is graphed against the rescue periodicity (reported in figures 5C and 6E). The data were fit with a linear regression (shown as a straight dashed line) and the 95% confidence interval is shown between the two curved dashed lines.

3) The authors are requested to provide a rationale why they chose CLOCK not BMAL1 (or CLOCK-BMAL1) to analyze the CRY1-interaction in the BiFC assay.

We chose to focus on CLOCK because of the many papers that suggest that BMAL1 is interacting with CRY through the CC helix and the C-terminal tail. The orientation of the tail and the secondary pocket suggested to us that the secondary pocket was a more likely interaction site for CLOCK rather than BMAL1. Our suspicions were confirmed by the

report from Michael et al. in PNAS in 2017. We have updated the text to try to make this rationale clearer (lines 609-16).

4) As for the CRY2 7m C1T and the CRY1 7m C2T, are there published studies suggesting the role of CRY c-terminal domain in the interaction with BMAL1-CLOCK? They should be mentioned in the discussion section.

Lines 694-698 cite the findings of these studies.

5) The authors may want to discuss the model of PER proposed in figure S9 based on the context-dependent action of PER. Serial studies from Sancer's group and others have proposed that PER can repress and de-repress the BMAL1-CLOCK through CRY1 (PMID: 25228643, 27688755, 21613214, 24794436). This model is conceptually similar to the model proposed in this study, but figure S9 premises the interaction of PER-CRY always leads to a transcription-repressive complex—which may not be always true.

Indeed, the model that we proposed does not square everything in the literature and we have updated the text from line 745 to 751 to reflect the work mentioned above.

6) Line 703. It would be fair to clearly mention that half-life independent period control by CRY is highlighted in recent studies (PMID: 28388406, 28017587), in particular, one of them (28388406) connects the half-life independent mechanism to the interaction between CRY1 and BMAL1/CLOCK.

We agree that these studies are in agreement with our findings. These studies were published after the original manuscript was prepared and lines 714-715 have been updated to cite them.

7) An "introduction" part from line 125 to line 181 is a bit verbose for the

result section. It can be summarized and/or part of this section can be moved to the introduction.

We have substantially rewritten this portion to be more succinct, and to clarify the goals of the SCA analysis.

Reviewer #2 (Remarks to the Author):

General evaluation:

The new findings described in this manuscript are very important to understand the mechanistic basis of CRY1 and CRY2 function in the circadian clock, specifically the role of the secondary pocket. Moreover, the manuscript provides essential information to explore the secondary pocket as a potential drug target to speed up the clock, e.g. to overcome jetlag. The presented work significantly advances our general mechanistic understanding of circadian clock operation, especially the role of protein-protein interactions in circadian regulation. The manuscript is of broad interest, including circadian research and the mechanistic understanding of cryptochrome/photolyase family members. The techniques are well described.

We thank the reviewer for these comments on the importance of the work.

Before publication, the following issues should be addressed:

General issue: The authors should be more careful using the term "allostic/allostery" (see detailed examples below). Also, the previously published data on the second pocket (Nangle et al , 2014; Michael et al, 2017, Czarna et al, 2013) should be presented more clearly to make it clearer what

was already know before and what is new in this manuscript (see detailed below).

We agree with the reviewer on both these points and have addressed these comments individually below under the “detailed issues” section.

Some detailed issues:

Abstract, Line 38: “weaker interactions – of CRY2 ? - with CLOCK/BMAL1” ? Please insert “of CRY2”, if this is what the author’s meant to say. As it is, this sentence is not clear.

The abstract has been updated to clarify this assertion.

Line 66 introduction “nevertheless it remains unclear which CRY interfaces lead to specific protein-protein interactions with CLOCK and BMAL1.” The authors place this statement in the introduction without acknowledging that a previous publication by Michael et al, PNAS 2017 already reported the importance of the CRY secondary pocket for Clock-PAS-B-HI loop interactions. Furthermore, several earlier publications e.g. by Chaves et al, 2006, Xu et al, 2013, Czarna et al, 2011 have already reported the role of the C-terminal coiled-coil helix of the CRY-PHR and the CRY-tail in transcriptional repression via the BMAL1-TAD.

It would be good if the authors would mention some of these earlier findings in the introduction, and not only in the discussion (Lines 686-691), to more clearly emphasize the new aspects of this manuscript.

Updated lines 68-85 to better emphasize the new aspects of this manuscript.

Line 160, p.8: In Photolyases, the second cofactor (e.g. MTHF) transfers energy to the FAD cofactor via Förster Resonance Energy Transfer. I would not call

this an "allosteric" modulation of the dynamics of the repair reaction, as the two chromophores and the DNA-lesion directly communicate with each other. Please remove "allosterically" in line 160.

Removed "allosterically".

Fig. 1C and Mat/meth p.38, line 831: please define "D_i". What does D_i = 1,2,3 .. mean in terms of conservation ? What does "weakly/moderately/more conserved " mean in Fig. 1C legend?

We have added additional language describing the calculation and interpretation of D_i in both the Fig 1C legend and the methods section. Along these lines, we also modified the methods section to further clarify the SCA calculations.

Fig. 1C legend: Please define "CPF" at first mention (it is defined later in the text)

Updated to define CPF in the figure legend.

Line 172: The dCRY structure presented in Zoltowski et al, Nature 2011 (Ref. 34) is not correct. The authors should cite Levy et al, Nature 2013 and Czarna et al, Cell 2013, which report correct dCRY structures.

These references have been updated.

Line 214: what do the authors mean by "the known allosteric mechanisms linking the two" (i.e. the FAD and secondary binding pocket) ? I would not call energy and electron transfer in the repair reaction of photolyases an allosteric mechanism (see above).

Line 261: see comment on line 214.

We agree that the repair reaction in the photolyases is not a true allosteric mechanism because it does not require coupled conformational change. Accordingly, we have edited the manuscript throughout to more precisely describe the relationship between the primary and secondary pockets and to be more careful with use of the term "allostery".

Fig. 3A: it may be helpful to show the "lower helix" as ribbon in addition to sticks.

Figure 3A has been updated with the "lower helix" backbone shown as ribbon and side chains as sticks.

Line 306: The authors should also mention the published results on R109Q regarding its reduced repression activity in circadian rescue experiments (Nagle et al, Elife 2014). This provides earlier evidence for the role of the secondary pocket in transcriptional regulation (see also line 130). It would be easier for the reader if earlier reported results on R109Q (Nagle et al, 2014; Michael et al, 2017), which were presented in the context of CRY1/2 crystal structures, would be placed in one location in the new manuscript.

This information was shifted to the intro (Lines 68-85) for clarity.

Line 337: looking at the alignments that the authors present I would not polarize the level of conservation between the upper and lower helix so much. There are exchanges in both motifs. So "largely conserved" vs. "highly divergent" is more contrasting than the alignments indicate.

We have tempered our description in the text accordingly (lines 339-341).

Line 687: I would not call a 1 to 10 μM affinity a "high affinity". High affinity

would rather imply a nM range affinity than a μ M range affinity.

The text has been edited to remove "high affinity".

Line 692: Are there nM range KD values available for the "high affinity interaction between CLOCK's HI loop and CRY at the secondary pocket" ? If not, I would not classify this interaction as "high affinity". If this interaction has not been quantified by determining KD values, it is dangerous to postulate, that the CLOCK-PAS-B-HI-loop/CRY-secondary pocket interaction recruits CRY primarily to the BMAL1/CLOCK complex and that the lower affinity (μ M) BMAL1-TAD-CRY-tail interaction is secondary- unless the authors have other evidence for this model. Also "allostery" (line 694) would imply an indirect conformational effect of the CLOCK-CRY interaction on the BMAL1-TAD-CRY-tail interaction. Is there evidence for this? If not I would rank it as a proximity effect rather than an allosteric effect.

We agree that there is no evidence yet to support our assertion that the interaction between the CRY secondary pocket and CLOCK PAS-B-HI-loop is high affinity. Likewise, we cannot support a claim that the ternary complex is formed through an indirect conformational change. Thus, the discussion has been edited to reflect this comment (lines 694-704).

Fig S5: The difference between Figure S5C and Fig. S5D/E is not clear. Please explain in Fig. legend.

The figure legend has been edited to clarify the difference.

Reviewer #3 (Remarks to the Author):

Rosensweig et al. present a study that is important in many aspects: (i) it provides a detailed molecular understanding of the role of cryptochromes in

the mammalian circadian clock; (ii) it sheds light on the evolution of cryptochromes with regard to their functional diversification.

Circadian clock regulation in mammals has still several mysteries, in particular with respect to cell-biological, but also with respect to structural aspects. Within the recent years, much has been learned about structural properties of cryptochromes also in complex with binding partners or small molecules regulating their stability. However, one puzzling aspect was largely unknown (at least from a structural point of view): Cry1 knockout leads to short periods, and Cry2 knockout to long periods (both in behaviour and cells). This study solves this puzzle by showing that the so-called secondary pocket in cryptochromes (i.e. residues therein) is divergent between CRY1 and CRY2 and modulates the binding strength to CLOCK thereby differentially inhibiting the transactivation activity of CLOCK/BMAL1.

In my opinion, this is a well-performed important study that eventually should be published. However, I have a couple concerns and suggestions the authors might want to consider.

We are grateful to this reviewer for the positive comments on our work.

1. SCA: (i) It has been known before that CLOCK binds to the secondary pocket of CRYs. 260 residues were identified to be sector residues – a very large fraction of the surface-exposed CRY residues. I don't really see a major benefit of the SCA for this particular study: the structures are known; residues of the secondary pocket could have been selected without SCA. At least, it would be helpful to see (in a table), how many residues of the secondary pocket are sector residues, how many and which of them have been mutated and tested and which ones resulted in an effect. I concede the evolutionary aspect is interesting, but may deserve a separate publication. In essence, do you really need SCA to study the importance of the secondary pocket? (ii) The presentation of SCA is lengthy (2 figures!) and sometimes difficult to understand for the non-expert reader – this should be improved.

The role of the SCA is two-fold. First, our analysis establishes that co-evolution between the primary and secondary pockets is a conserved feature of both CRYs and PHLs, consistent with the observation that the two pockets are functionally coupled in both families. Secondly, it provides framework for guiding (and interpreting) our mutagenesis results. At the time that the SCA was undertaken, it was not yet definitively known whether CLOCK was binding to the secondary pocket. We focused on the secondary pocket because so much of this region was highlighted as part of the coevolutionary network in the SCA that it suggested a major role in CRY function. We have edited the main text and methods to clarify the presentation (and interpretation) of SCA.

2. Allostery: (i) The functional sites of mammalian CRYs should be better explained in the introduction, e.g. what is the flavin pocket doing in mammals? Why is this the active site? (ii) It is unclear, how mutations at the secondary pocket influence the "active" site. It is obvious that they modulate binding affinity to CLOCK, but it is unclear whether affinity to BMAL1-TAD is allosterically regulated. Is it really allosteric regulation in mammalian CRYs or is this a "historic" term referring to pterin's role of photon capture and transmitting energy to flavin in the photolyase's active site?

We agree that our use of the term "allostery" is imprecise. More specifically, we mean to say that: 1) in the photolyases the two pockets are functionally linked via photon capture at the secondary pocket and energetic transfer to the flavin in the active site (primary pocket), 2) the SCA shows that the two pockets co-evolve across both the PHLs and CRYs, 3) this suggests that the two sites are also functionally linked in the CRYs (though not necessarily through allosteric regulation mediated by coupled conformational change). Our experimental data indicate that binding between CRY/PER (at the primary pocket) and

CRY/CLOCK/BMAL1(at the secondary pocket) is greatly enhanced when all four components are expressed together, however they do not establish if the enhancement is allosteric in nature. We have edited the manuscript throughout to be clearer about the proposed relationship between the two pockets, and the interpretation of the SCA.

3. The authors should better discuss or even might want to test the role of CRY stability in this context. E.g. can a mutant with very weak CLOCK binding properties (e.g. D38A) be rescued by Fbxl3 knockdown? Would Fbxl3 knockdown have a differential effect for Cry1 and Cry2 knockouts?

We agree that this is a worthwhile experiment to perform. However, the lead author has transitioned to a new postdoctoral position and cannot perform these experiments. We have added a note in the discussion (lines 715-720) to address this possibility.

4. Overall, there seems to be a relation between rescue period, repression activity and binding to CLOCK. Sometimes, however, this does not apply (Figure 5B,C and Figure 3E,F). What are the other determinants? E.g. are K107 and E108 mutants more stable? In general, the authors use primarily three types of assays: (i) rescue, which provides period and repression; (ii) binding assays with IP and (iii) stability measurements. It is unclear, why these assays are not always used for all mutants tested to get a comprehensive picture.

To perform all three assays requires that mutagenesis be performed on three different vectors, which became combinatorially complex as the group of mutants that we tested grew. Though not all the data are shown here, we ultimately tested over 80 different mutants in the rescue assay and a large subset of these mutants were tested in our binding assay experiments. For measurement of stability, we focused primarily on mutants that had large effects on periodicity in order to rule out

destabilizing effects. Though we have not performed stability assays for all the mutants shown in this paper, we can address the question of whether the K107 mutant is more stable based on the literature. Yoo et al. Cell 2013 demonstrated that K107 is a target for ubiquitination by FBXL3 and serves as a degron. Thus, we might expect that its longer period in the rescue assay stems from this reported role in CRY stability.

5. The bimolecular fluorescence complementation assay is elegant and should be further exploited to test, whether PERs indeed strengthen the binding of CRY2 to CLOCK.

We attempted this experiment, but saw no changes. However, it should be noted that BiFC can require substantial construct engineering to bring fluorescent protein fragments into an appropriate orientation to bind and fluoresce. We were forced to engineer CLOCK to delete some N-terminal regions and a large C-terminal region beyond the PAS-B domain (including the exon 19 region) in order to observe interactions with CRYs. It is not yet clear how PER stabilizes the interaction between CRY and CLOCK/BMAL1, though it has been reported that CLOCK's exon 19 region may interact with PER (Lee et al. PNAS 2016). Thus, it is impossible to definitively conclude whether or not PER stabilizes the ternary complex from this experiment.

6. Figure 8: Does the tails modulate binding to CLOCK/BMAL1?

We have not run any experiments to address this question with our own data, but experiments from the literature support the conclusion that the tails do in fact modulate binding to CLOCK/BMAL1. Experiments from Czarna et al. JBC 2011, Cell 2013, Xu et al. NSMB 2014, and Patke et al. Cell 2017 all suggest that the CC helix and tail play a role in the

interaction between CRY and BMAL1. We have added a discussion of these data from lines 694 to 715.

REVIEWERS' COMMENTS:

Reviewer #1 (Remarks to the Author):

The authors have addressed all my comments. This paper is now ready for the publication. The correlation plot shown in the rebuttal letter is a good representation showing that CRY-CLOCK-BMAL1 affinity explains the period length at least in part. Thus, upon the authors' decision, it is welcomed to include these plots for the final version of this manuscript, although I agree that nature of co-IP experiments may limit the statistical interpretation.